# UFL: Uncertainty-Driven Federated Learning

## Abstract

Federated Learning (FL), a privacy-preserving distributed machine learning, encounters numerous challenges in practical applications, notably Data Heterogeneity (DH). Current methods primarily address DH, relying on coarse dataset statistics, server aggregation, or local model uncertainty. This paper reveals that FL exhibits a distinct sample-level uncertainty distribution during training, characterized by a pronounced long-tail effect. We further show that this long-tail effect is not solely attributable to DH, but is also an inherent characteristic of the FL framework itself. To this end, we propose **U**ncertainty-driven **F**ederated **L**earning (**UFL**), a framework designed to address the uncertainty challenge at the sample level. UFL employs Monte Carlo (MC) dropout to estimate sample uncertainty and adaptively re-weights the loss function accordingly. Moreover, we design U-Agg, a robust aggregation method using clients' accumulated high-uncertainty sample uncertainty to adjust aggregating weights and improve convergence with theoretical guarantees. Unlike existing approaches that alleviate DH at coarser levels, UFL introduces a sample-centric perspective that directly addresses the uncertainty challenge from its fundamental source, offering an orthogonal yet complementary dimension to traditional techniques. Extensive experiments demonstrate that UFL outperforms SOTA FL methods by mitigating the long-tail effect of sample uncertainty, offering a novel and complementary perspective on sample-level uncertainty to enhance FL efficacy over DH solutions.

## 1 Introduction

Federated Learning (FL) (McMahan et al., 2017) is a distributed machine learning paradigm that enables multiple clients to collaboratively train a global model without sharing their raw data. This approach preserves data privacy and security, making it particularly suitable for applications in sensitive domains such as healthcare, finance, and autonomous driving (Fu et al., 2024; Chen et al., 2021; 2024; Bai et al., 2024; Guo et al., 2024c). However, the performance of FL is often hindered by Data Heterogeneity (DH) (Karimireddy et al., 2020), where clients' local datasets exhibit significant variations in distribution and volume. This heterogeneity can lead to suboptimal model performance and convergence issues. To this end, existing works focus on enhancing local training (Li et al., 2020; Karimireddy et al., 2020; Wang et al., 2021; Zhang et al., 2023), utilizing gradients or adding regularization (Jin et al., 2023; Li et al., 2021; Bergström et al., 2024; Chen & Vikalo, 2024; Tan et al., 2024), employing knowledge distillation (Park et al., 2023; Zhang et al., 2024b), and constructing proxy datasets for auxiliary training (Park et al., 2023; Huang et al., 2024), or designing robust aggregation method based on model parameter similarity or local statistical properties (Blanchard et al., 2017; Sun et al., 2019; Ghosh et al., 2019; Ye et al., 2023b).

Another direction has been a growing interest in addressing the DH issue by estimating the model uncertainty in FL. One well-known approach leverages mutual information (MI), which has evolved from its traditional role in establishing generalization bounds in centralized and federated settings (Xu & Raginsky, 2017; Asadi et al., 2018; Barnes et al., 2022; Yagli et al., 2020) to more direct applications in FL. For example, FedMDMI (Zhang et al., 2024a) employs MI-based regularization to alleviate local posterior biases induced by DH. In parallel, Bayesian FL provides a principled framework for refining local posterior estimates (Al-Shedivat et al., 2021), reducing their biases (Guo et al., 2023), and aggregating or distilling distributed knowledge in a robust manner to construct a coherent global model (Chen & Chao, 2020; Bhatt et al., 2023; Plassier et al., 2023). A key advantage of Bayesian methods lies in their ability to quantify epistemic uncertainty at the model level, thereby

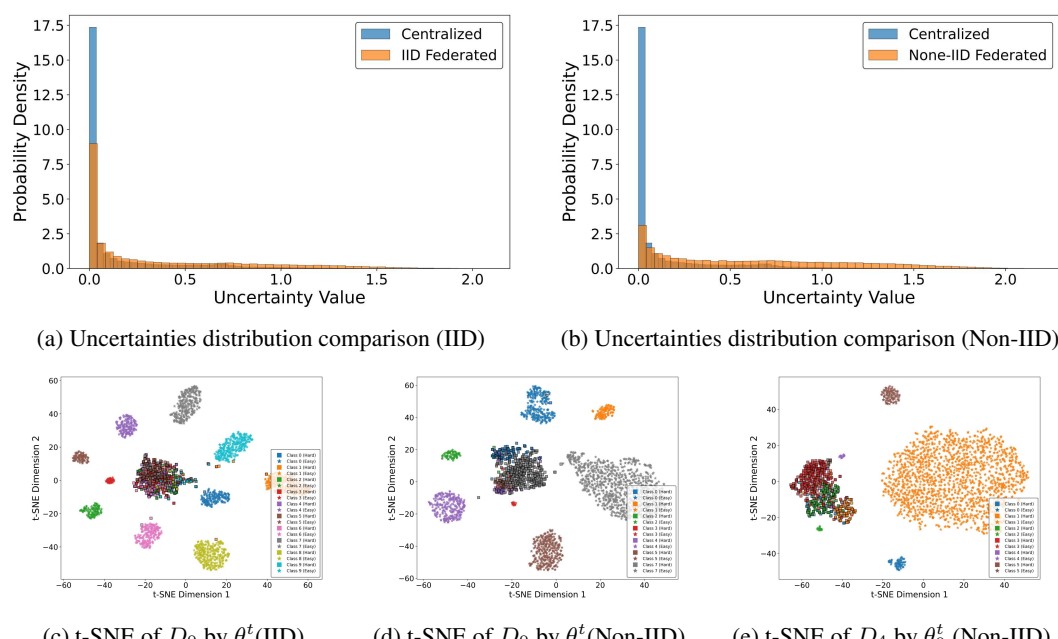

(a) Uncertainties distribution comparison (IID)      (b) Uncertainties distribution comparison (Non-IID)

(c) t-SNE of $D_0$ by $\theta^t$(IID).     (d) t-SNE of $D_0$ by $\theta^t$(Non-IID).     (e) t-SNE of $D_4$ by $\theta_0^t$ (Non-IID).

Figure 1: **(1) The long-tail effect of sample uncertainty**. (a) and (b) compare sample uncertainty distributions of centralized training versus FL under IID and Non-IID settings at 75% epochs. **(2) Feature inseparability of high-uncertainty samples, transcending data distribution.** $\theta^t$, global model; $\theta_k^t$, local models; $\mathcal{D}_k$, local dataset of client $k$. (See Appendix E for details.)

offering a principled approach to counteract DH effects in FL. However, these approaches primarily focus on estimating model uncertainty, an inherently coarse-grained perspective that may not fully capture the heterogeneity that exists at the data level. This limitation motivates a central research question: *can we develop fine-grained measures of DH that go beyond characterizations restricted to local datasets, local updates, or model-level uncertainty?*

In this paper, we address the DH effect in FL by explicitly estimating sample-level uncertainty. Compared with centralized training, FL inherently amplifies sample uncertainty (Figure 1a). When the DH effect is further worsened by Non-IID data partitioning (Figure 1b), the resulting uncertainty exhibits a distinct long-tail distribution. Moreover, high-uncertainty samples are typically entangled in the feature space, as illustrated in Figure 1c–e. This high uncertainty phenomenon is not only attributable to DH among clients, but also intrinsic to the FL paradigm itself—an important aspect that has remained largely unexplored in existing literature.

To this end, we propose **U**ncertainty-driven **F**ederated **L**earning (**UFL**), a novel framework that addresses the high uncertainty challenges posed by FL. Instead of constructing macroscopic DH mitigation strategies, UFL approximates the sample-level uncertainty via Monte Carlo (MC) Dropout (Gal & Ghahramani, 2016), and dynamically reweights the local loss function according to the sample uncertainty, thus improving generalization of the local model. We further propose a robust aggregation method called uncertainty aggregation (U-Agg) for the server side by utilizing the accumulated uncertainty from local models with theoretically guaranteed robustness. These two components work synergistically to create a closed-loop methodology that addresses the intrinsic challenges posed by high-uncertainty issues in FL at both client and server levels. In summary, our contributions can be summarized as

- This paper reveals and analyzes the under-explored sample-level uncertainty issue in FL, demonstrating the natural phenomenon of high uncertainty compared to centralized training. This suggests that FL not only exacerbates DH issues but also introduces additional challenges due to the decentralized training process.

- We propose the Uncertainty-driven Federated Learning (UFL), which utilizes MC Dropout to identify sample-level uncertainty during local training and adaptively re-weights the local loss function, thus improving the generalization of the local model.

- We propose U-Agg, an uncertainty-aware robust aggregation mechanism, and provide theoretical grounding for its design. Our analysis demonstrates an inverse relationship between client uncertainty and optimal aggregation weights, a principle directly embodied by U-Agg's strategy of prioritizing clients more proficient with high-uncertainty samples.

- Comprehensive empirical validation shows UFL outperforms current DH solutions (using local dataset statistical features, proxy datasets, or model similarity), highlighting the novel and DH-complementary perspective as a key avenue for enhancing FL performance.

## 2  RELATED WORKS

### 2.1  ENHANCING LOCAL TRAINING FOR DH

(1) **Knowledge Distillation (KD)**: FedDefender (Park et al., 2023) first distills knowledge from the global model to local models and then applies meta-learning with a nosing dataset. Building on KD, FedGMKD (Zhang et al., 2024b) integrates graph-based multi-KD with prototype-driven methods to strengthen client–server collaboration, while FedKF (Zhou et al., 2023) introduces a lightweight generator that ensembles user information to provide inductive bias. Along the same line, Guo et al. (Guo et al., 2024a) develop a selective KD framework to better handle DH. **(2) Proxy dataset**: SDEA (Huang et al., 2024) adopts self-driven entropy aggregation by testing models on a random public dataset to inform decision-making, while Sageflow (Park et al., 2021) leverages predictive entropy on proxy data to evaluate each model's contribution. **(3) Local dataset or gradient information**: Many approaches address DH in FL by leveraging local data and gradient signals. Some stabilize optimization, such as FedProx with proximal regularization (Li et al., 2020) and SCAFFOLD with control variates (Karimireddy et al., 2020). Others adapt training dynamics, including FedNova with task allocation and pruning (Wang et al., 2021) and DBE with personalized vectors (Zhang et al., 2023). A further line aligns representations, where FedDyn uses update trajectory differences (Jin et al., 2023), MOON anchors features to a global benchmark (Li et al., 2021), FedFM unifies client-specific "feature dialects" (Ye et al., 2023a), and FedRep learns shared representations with local heads (Collins et al., 2021). Unlike existing approaches, UFL tackles DH through a more fine-grained lens by quantifying sample-level uncertainty to strengthen local training and enabling robust server aggregation.

### 2.2  ROBUST AGGREGATIONS FOR DH

Existing robust aggregation approaches typically rely on coarse-grained DH metrics, such as clustering clients with similar feature distributions (Guo et al., 2024b; Long et al., 2023; Zhu et al., 2023), ranking or filtering updates based on gradient statistics (Blanchard et al., 2017; Ghosh et al., 2019; Tian et al., 2024), and weighting contributions using structural priors such as class hierarchies (Ye et al., 2023b; Lucchetti et al., 2022; Bhatt et al., 2023). While effective to some extent, these methods rely on coarse client-level metrics communicated to the server, assuming such signals are sufficient to capture heterogeneity. In contrast, UFL quantifies sample-level uncertainty, enhancing local learning while guiding robust aggregation, thereby capturing fine-grained patterns beyond the reach of client-level methods.

### 2.3  BAYESIAN PERSPECTIVE FOR DH

Bayesian uncertainty quantification (UQ) has recently emerged as a powerful tool for enhancing robustness to distribution shift (Li et al., 2024; Suk & Kim; Ovadia et al., 2019; Ritter et al., 2021; Zhang et al., 2024c; Qu et al., 2024), guiding exploration in reinforcement learning (Blundell et al., 2015; Lockwood & Si, 2022; Parisi et al., 2024; Gal & Ghahramani, 2016), and reducing annotation costs in active learning (Werner & Schmidt-Thieme, 2025; Woo, 2023; Settles, 2009). In FL, mitigating distributional heterogeneity from a Bayesian perspective has followed two main directions: regulating model–data MI, which links data complexity to generalization and has been extended to FL with MI-based bounds and regularizers such as FedMDMI (Xu & Raginsky, 2017; Asadi et al., 2018; Barnes et al., 2022; Yagli et al., 2020; Zhang et al., 2024a); and Bayesian inference, where methods including FedPA, FedEP, FedBE, Bhatt et al., and VR-FALD (Al-Shedivat et al., 2021; Guo et al., 2023; Chen & Chao, 2020; Bhatt et al., 2023; Plassier et al., 2023) approximate or aggregate local posteriors to correct bias. Yet these approaches focus largely on parameter uncertainty, overlooking

the distinct long-tail distribution of sample-level uncertainty we observe in FL (Fig. 1), where high-uncertainty samples remain indistinguishable in feature space. We therefore propose tackling DH by directly modeling sample uncertainty, offering a more effective alternative to parameter-centric approaches.

## 3 UFL: UNCERTAINTY-DRIVEN FEDERATED LEARNING

Inspired by empirical findings and the limitations of existing work, we propose Uncertainty-driven Federated Learning (UFL). UFL consists of: (1) local training with sample uncertainty estimation, and (2) uncertainty-based aggregation. The graphical illustration of the UFL diagram and the overall FL framework can be found in Appendix B and Appendix D.

**Uncertainty Estimation and Local Training**   Each local node receives the global model $\theta_k$ from the server and trains it on its local dataset. Before training, $\theta_k$ is used to compute the uncertainty for each sample. Specifically, all local training data is fed into $\theta_k$ (i.e., the model aggregated in the previous round). Multiple forward passes are performed using MC dropout (Gal & Ghahramani, 2016) to obtain the average prediction distribution $\bar{p}$ for each sample. Then the entropy can be computed to represent the uncertainty $u_i$ for $i$-th sample in the local dataset $\mathcal{D}_k$:

$$u_i = -\sum_{c=1}^{C} y_{i,c} \log (\bar{p}_{i,c}), \tag{1}$$

where $C$ is the total number of classes, $y_{i,c}$ is the one-hot encoding of the true probability distribution for the $c$-th class of sample $i$ and $\bar{p}_{i,c}$ denotes the average prediction probability for the $c$-th class of sample $i$ obtained via multiple MC dropout passes from $\theta^t$.

The uncertainties $u_i$ are sorted in descending to form a set $\mathcal{U} := \{u_1, u_2, \ldots, u_{|\mathcal{D}_k|}\}$. We let $\mathcal{I}$ be the top-$m$ subset of $\mathcal{U}$, i.e., $\mathcal{I} \in \arg\max_{J \subseteq \{1,2,\cdots,|\mathcal{D}_k|\}, |J|=m} \sum_{j \in J} u_j$. Thus, samples in $\mathcal{I}$ can be considered as the higher uncertainty. During training, all samples are used, but high-uncertainty samples will be assigned higher weights to amplify their influence on the loss. Specifically, an $\alpha > 0$ scaling factor is applied to $u_i$ to construct the re-weighted vector $\boldsymbol{\lambda}$:

$$\lambda_i = \begin{cases} 1 + \alpha u_i, & \text{if } i \in \mathcal{I}, \\ 1, & \text{otherwise.} \end{cases} \tag{2}$$

In practice, we usually let $|\mathcal{I}|$ be $5\% \sim 30\%$ of $|\mathcal{D}_k|$. Consequently, the $|\mathcal{I}|$ high-uncertainty samples are non-contiguous in the local dataset, and the $\alpha$ scaling is applied based on their indices. For low-uncertainty samples, the weight is set to $1.0$, leaving their contribution to the cross-entropy loss unchanged.

During local training, weights are assigned based on sample indices and applied to individual sample losses. The re-weighted losses are then averaged to prioritize the contribution of high-uncertainty samples. The overall loss for $B$ samples is given by:

$$\mathcal{L} = -\frac{1}{B} \sum_{i}^{B} \lambda_i \sum_{c=1}^{C} y_{i,c} \log (p_{i,c}), \tag{3}$$

where $p_{i,c}$ denotes the probability of the $i$-th sample belonging to class $c$, $\theta_k$ is local model.

**Uncertainty-Based Aggregation (U-Agg)**   This paper introduces an adaptive weighting method based on uncertainty quantification. During local training, client $k$ sums the uncertainty of its $|\mathcal{I}|$ high-uncertainty samples to compute the local uncertainty aggregate, $U_k = \sum_{i=1}^{|\mathcal{I}|} u_i$, and uploads it to the server. The server normalizes the uncertainty values across all clients to obtain $\hat{U}_k$. Relative aggregation weights are then computed as $1 - \hat{U}_k$. These weights are further normalized to ensure their sum equals 1, yielding the final weight vector $\widetilde{\mathbf{U}} = (\widetilde{U}_1, \widetilde{U}_2, \ldots, \widetilde{U}_k)$. Finally, the server performs a weighted average of the model parameters uploaded by all clients based on these weights, producing the updated global model for the next training round. The proposed U-Agg is detailed as follows:

$$\theta^{t+1} = \sum_{k=1}^{K} \widetilde{U}_k \theta_k^{t+1} = \sum_{k=1}^{K} \frac{1 - \hat{U}_k}{\sum_{i=1}^{K} (1 - \hat{U}_k)} \theta_k^{t+1}, \tag{4}$$

where $\widetilde{U}_k \in (0, 1)$ represents the normalized cumulative uncertainty of the $k$-th client.

**Discussion**    UFL addresses sample-level uncertainty challenges in FL, including the long-tail effect and the persistent difficulties in learning high-uncertainty samples. The local training component, by adaptively increasing loss weights for high-uncertainty samples, a principle with parallels to Focal Loss (Lin et al., 2018), encourages models to prioritize these challenging instances. Complementing this, the U-Agg mechanism leverages client-level high-uncertainty sample uncertainty to modulate aggregation weights. This approach reduces the influence of clients that struggle with difficult local data, thereby curtailing noise propagation and fostering a more robust and generalizable global model. UFL thus offers a sample-centric perspective that can complement traditional aggregation and DH mitigation strategies.

## 4    THEORETICAL ANALYSIS ON CONVERGENCE AND AGGREGATION

This section provides a principled analysis of **UFL**, establishing theoretical grounding for the proposed uncertainty-aware mechanisms as well as verifying the correctness of the U-Agg aggregation approach. We adopt a convergence analysis framework commonly used in FL (Li et al., 2020; Karimireddy et al., 2020; Ye et al., 2023b). Our analysis demonstrates that accounting for client uncertainty leads to a convergence bound that justifies the U-Agg weighting scheme. (Detailed FL diagram in Appendix D)

**Setup and Uncertainty Measure**    We consider an FL system with $K$ clients indexed by $k \in \{1, \ldots, K\}$. The goal of FL is to minimize the global empirical risk function:

$$F(\theta) = \sum_{k=1}^{K} p_k F_k(\theta_k), F_k(\theta_k) = \frac{1}{|\mathcal{D}_k|} \sum_{(x_i, y_i) \in \mathcal{D}_k} l(x_i, y_i; \theta_k), \qquad (5)$$

where $F_k(\theta_k)$ is the local empirical risk on client $k$. The global objective $F(\theta)$ is typically defined with $p_k = |\mathcal{D}_k|/|\mathcal{D}|$, but we will analyze the convergence of algorithms using general aggregation weights $p_k$ (where $\sum p_k = 1, p_k \geq 0$) in the update step.

### 4.1    ASSUMPTIONS FOR CONVERGENCE ANALYSIS

Our analysis relies on the following standard assumptions in FL literature (Li et al., 2020; Karimireddy et al., 2020; Ye et al., 2023b):

**Assumption 4.1** (Smoothness)**.** The local loss functions $F_k(\theta_k)$ are $L$-smooth for all clients $k$. Consequently, the global loss function $F(\theta) = \sum_{k=1}^{K} p_k F_k(\theta_k)$ is also $L$-smooth. That is, there exists a constant $L \geq 0$ such that for all clients $k$ and any parameters $\theta, \theta'$:

$$||\nabla F_k(\theta_k) - \nabla F_k(\theta'_k)|| \leq L||\theta_k - \theta'_k||, \quad \forall \theta_k, \theta'_k, k. \qquad (6)$$

**Assumption 4.2** (Bounded Below)**.** The global loss function $F(\theta)$ is bounded below by $F_{inf}$.

$$F(\theta) \geq F_{inf}, \forall \theta. \qquad (7)$$

**Assumption 4.3** (Stochastic Gradient Properties)**.** The stochastic gradient $g_k(\theta; \xi)$ computed on a mini-batch $\xi$ from $_k$ is an unbiased estimate of the true local gradient $\nabla F_k(\theta_k)$, i.e., $\mathbb{E}_\xi[g_k(\theta; \xi)] = \nabla F_k(\theta_k)$, and has bounded variance $\sigma$:

$$\mathbb{E}_\xi[||g_k(\theta; \xi) - \nabla F_k(\theta_k)||^2] \leq \sigma^2. \qquad (8)$$

To capture the impact of uncertainty-driven heterogeneity, we replace the original assumption on gradient norms with one based on gradient divergence, linking it to our uncertainty measure:

**Assumption 4.4** (Uncertainty-Modulated Gradient Divergence, Rationale in Appendix F)**.** The expected squared divergence between a local gradient and the global average gradient $\nabla F(\theta) = \sum_{j=1}^{K} \frac{|D_j|}{|\mathcal{D}|} \nabla F_j(\theta_j)$ is bounded, and this bound is influenced by the client's relative uncertainty. Specifically, there exist non-negative constants $B_G^2$ and $V$ such that for all clients $k$ and $\theta$:

$$\mathbb{E}||\nabla F_k(\theta_k) - \nabla F(\theta)||^2 \leq B_G^2 + V\hat{U}_k. \qquad (9)$$

## 4.2 CONVERGENCE ANALYSIS AND OPTIMAL AGGREGATION WEIGHTS

Based on these assumptions, we analyze the convergence of a generalized FedAvg algorithm where clients perform $\tau$ local SGD steps with learning rate $\eta$ starting from $\theta^t$, producing local models $\theta_k^{t+1}$, and the server aggregates these using potentially arbitrary weights $p_k$ to form $\theta^{t+1} = \sum_{k=1}^{K} p_k \theta_k^{t+1}$. For simplicity in stating the bound, we consider the update $\theta^{t+1} = \theta^t - \eta_{eff} \sum p_k \Delta_k^t$, where $\Delta_k^t$ is the effective update direction from client $k$ over $\tau$ steps. The $\eta_{eff}$ is an effective learning rate, typically set to 1 for model aggregation or $\eta$ for gradient aggregation. The analysis focuses on bounding the expected squared norm of the global gradient, using $F$ defined with $|\mathcal{D}_k|/|\mathcal{D}|$.

**Theorem 4.5** (Convergence Bound with Uncertainty-Modulated Divergence (See Appendix G for details)). *Under Assumptions 4.1-4.4, for a sufficiently small effective learning rate $\eta_{eff}$ (or local $\eta$) and $\tau$ local steps ($\tau > 1$), the average expected squared gradient norm over $T$ rounds is bounded. A simplified structure of the bound, highlighting key dependencies, is:*

$$
\frac{1}{T} \sum_{t=0}^{T-1} \mathbb{E}||\nabla F(\theta^t)||^2 \leq \underbrace{\frac{C_1(F(\theta^0) - F_{\inf})}{T\eta_{eff}}}_{\text{Term 1: Initial Error}} + \underbrace{C_2 \eta_{eff}\sigma^2 \left( \sum_{k=1}^{K} \frac{p_k^2}{|\mathcal{D}_k|/|\mathcal{D}|} \right)}_{\text{Term 2: Local SGD Variance}} + \underbrace{C_3 \eta_{eff}(\tau-1)\sigma^2}_{\text{Term 3: Local Steps Variance}}
$$

$$
+ \underbrace{C_4 \eta_{eff}\tau \sum_{k=1}^{K} p_k(B_G^2 + V\hat{U}_k)}_{\text{Term 4: Heterogeneity \& Uncertainty}} \quad \text{(from divergence)} + \underbrace{C_5 \eta_{eff}||\nabla F(\theta^t)||^2 \sum_{k=1}^{K} (p_k - |\mathcal{D}_k|/|\mathcal{D}|)^2}_{\text{Term 5: Weight Mismatch Effect (Approx.)}},
$$
(10)

*where $C_1, \ldots, C_5$ are positive constants depending on $L, \sigma^2, \tau$, but independent of $T, p_k, \hat{U}_k$. The term $C_4 \eta_{eff}\tau \sum p_k(B_G^2 + V\hat{U}_k)$ arises from bounding the effects of gradient divergence using Assumption 4.4.*

Theorem 4.5 highlights that convergence is affected by stochastic gradient variance $\sigma^2$, the number of local steps $\tau$, the choice of aggregation weights $p_k$, and the uncertainty-modulated gradient divergence $B_G^2 + D\hat{U}_k$. The term $C_4 \eta_{eff}\tau V \sum p_k \hat{U}_k$ represents the penalty introduced by client uncertainty via its contribution to gradient divergence. To accelerate convergence (minimize the bound), we seek aggregation weights $p_k^*$ that minimize the terms dependent on $p_k$.

**Proposition 4.6** (Optimal Aggregation Weight Form). *Minimizing the dominant terms related to $p_k$ in the convergence bound equation 10, such as $C_2 \eta_{eff}\sigma^2 \sum \frac{p_k^2}{|\mathcal{D}_k|/|\mathcal{D}|} + C_4 \eta_{eff}\tau \sum p_k(B_G^2 + V\hat{U}_k)$, subject to the constraints $\sum p_k = 1, p_k \geq 0$, leads to optimal weights $p_k^*$ that are inversely related to the client's uncertainty $\hat{U}_k$. A conceptual form is: $p_k^* \propto \omega_k \times (\kappa - \zeta \times \hat{U}_k)$, where $\omega_k$ represents baseline importance, e.g., related to $|\mathcal{D}_k|/|\mathcal{D}|$ from the variance term, and the crucial feature is the negative dependence on $\hat{U}_k$. $\kappa$ is a constant arising from normalization and baseline terms, and $\zeta$ is a positive constant ($\zeta > 0$) reflecting the negative impact of uncertainty $\hat{U}_k$.*

Proposition 4.6 provides theoretical validation for the U-Agg mechanism. It demonstrates that, to minimize a convergence bound influenced by uncertainty-modulated gradient divergence, the aggregation weights $p_k^*$ should indeed be reduced for clients exhibiting higher uncertainty levels ($\hat{U}_k$). This aligns directly with the principle of U-Agg, where weights $\widetilde{U}_k$ are calculated based on $(1 - \hat{U}_k)$, effectively implementing this inverse relationship. (See Appendix J and H for details)

## 5 NUMERICAL RESULTS

The experimental design of this study comprises three parts: (1) Primary experiments compared UFL with SOTA FL methods. (2) Aggregation experiments compared U-Agg with SOTA aggregating methods. (3) Ablation experiments investigated hyperparameter impacts on the global model. (4) Exploring uncertainty trends during training.

## 5.1 Experimental Settings

**Dataset**: This study conducts experiments on common datasets with $C = 10$ classes (e.g., MNIST (Deng, 2012), SVHN (Netzer et al., 2011), Fashion-MNIST (Xiao et al., 2017), CIFAR-10 (Krizhevsky & Hinton, 2009)) and extends to more complex datasets with $C \geq 100$ classes. Unlike existing studies that focus solely on 10-class scenarios (Huang et al., 2024; Ye et al., 2023b; Zhang et al., 2024b), etc., we further evaluate our method on CIFAR-100 (Krizhevsky & Hinton, 2009) and TinyImageNet-200 (mnmoustafa & Ali, 2017). **FL Setup**: The datasets are partitioned Non-IID following a Dirichlet distribution, with a hyperparameter $\beta = 0.5$ controlling the degree of DH. The FL training includes 10 clients performing 100 rounds of global aggregation and 1 epoch local training for each round. Each client trains with a batch size of 64. ResNet18 (He et al., 2016) is deployed on all clients and used to train all datasets. The learning rate is set to 0.01, and the SGD algorithm is employed to update local models.

**Baseline**: FedAvg (McMahan et al., 2017) serves as the baseline for all experiments. The comparison includes two categories of SOTA methods: Local training improving FL (Scaffold (Karimireddy et al., 2020), FedDefender (Park et al., 2023), FedProx (Li et al., 2020), and FedNova (Wang et al., 2021)) and aggregation algorithms(Krum (Blanchard et al., 2017), Norm Bound (Sun et al., 2019), and FedDisco (Ye et al., 2023b) (which incorporates local enhancement)). **MC Dropout Parameters**: The global model performs $N = 10$ forward passes to estimate the average prediction probabilities for each sample, which are used to compute uncertainty as a measure of difficulty. By default, we let $|\mathcal{I}|/|\mathcal{D}_k| = 30\%$, i.e., top-30% samples are regarded as high-uncertainty samples. The weight scaling factor $\alpha$ is set to 0.2. These parameters will be further explored in the ablation study.

Table 1: Comparison of accuracy for local enhanced FL methods under the average aggregation on six datasets (%).

| Method | CIFAR-10 | MNIST | Fashion-MNIST | SVHN | CIFAR-100 | TinyImageNet-200 |
|---|---|---|---|---|---|---|
| FedAvg | 86.02 (↑2.25) | 99.03 (↑0.12) | 91.82 (↑0.67) | 92.95 (↑2.05) | 62.44 (↑3.71) | 48.16 (↑3.80) |
| Scaffold | 87.60 (↑0.67) | 99.10 (↑0.05) | 91.71 (↑0.78) | 93.60 (↑1.40) | 55.54 (↑10.61) | 47.61 (↑4.35) |
| FedDefender | 87.95 (↑0.32) | 99.02 (↑0.13) | 92.49 (↑0.00) | 94.55 (↑0.45) | 65.44 (↑0.71) | 49.84 (↑2.12) |
| FedProx | 86.44 (↑1.83) | 99.01 (↑0.14) | 91.83 (↑0.66) | 94.10 (↑0.90) | 62.56 (↑3.59) | 48.13 (↑3.83) |
| FedNova | 85.68 (↑2.59) | 98.97 (↑0.18) | 91.70 (↑0.79) | 94.10 (↑0.90) | 62.18 (↑3.97) | 49.08 (↑2.88) |
| UFL (+Average) | **88.27** | **99.15** | **92.49** | **95.00** | **66.15** | **51.96** |

*Note.* ↑ indicates that our method *improves* upon the baseline, while ↓ indicates a performance decrease.

## 5.2 Main Results

**Effectiveness of UFL Local Training**   To verify the effectiveness of UFL-enhanced local training, we combine the proposed UFL local training and average aggregation for the sake of a fairness comparison. Table 1 shows the classification accuracy of baselines and our method on six datasets. Specifically, on CIFAR-10, UFL reached an accuracy of 88.27%, outperforming FedAvg (McMahan et al., 2017) and FedDefender (Park et al., 2023) by 2.25% and 0.32%. On MNIST and Fashion-MNIST, UFL achieved 99.15% and 92.49% accuracy, surpassing existing methods. Moreover, UFL demonstrates advantages, particularly on CIFAR-100 and TinyImageNet-200. For instance, UFL achieves an accuracy of 66.15% on CIFAR-100, surpassing the best existing method, FedDefender, by 0.71%. On TinyImageNet-200, UFL attains 51.96% accuracy, outperforming FedAvg and FedDefender by 3.80% and 2.12%. These results highlight the effectiveness and critical role of the proposed UFL reweighted adjustment in local training.

**Mitigation of Long Tail Effect**   This experiment compares the sample uncertainty probability density distributions of UFL against other methods (Karimireddy et al., 2020; Park et al., 2023; Li et al., 2020; Wang et al., 2021; Blanchard et al., 2017; Sun et al., 2019; Ye et al., 2023b) under a Non-IID setting of CIFAR-10 and CIFAR-100. As shown in Figure 2a and 2b, UFL's sample uncertainty is concentrated in the low-value region near 0 and decays rapidly. UFL's distribution in the high-uncertainty areas is smaller than that of other methods, exhibiting the shorter "tail", which implies that most samples are classified with high confidence. As depicted in Figure 2b, SCAFFOLD continues to exhibit a pronounced long-tail effect on the CIFAR-100 dataset after training. In Figure 2a, the sample uncertainty for UFL is predominantly concentrated in the range of 0 to 0.25, indicating

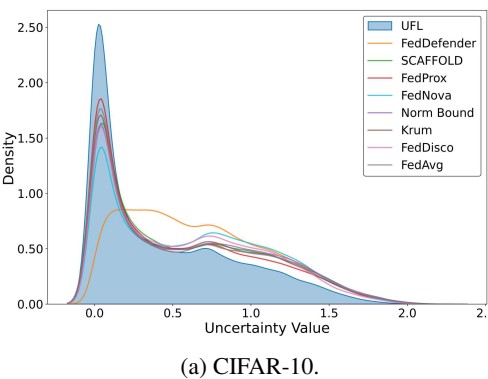
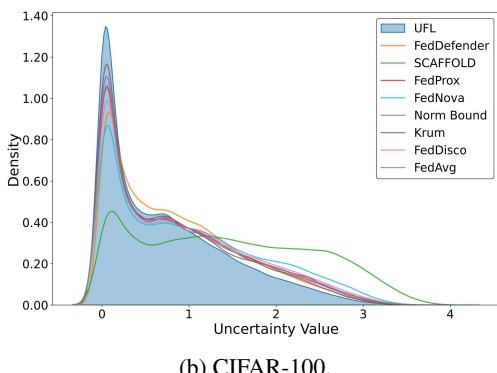

(a) CIFAR-10.               (b) CIFAR-100.

Figure 2: A comparison of the long tail effect. (a) The distribution of uncertainty value on CIFAR 10. (b). The distribution of uncertainty value on CIFAR 100.

that the model assigns classifications with high certainty. Conversely, methods leveraging dataset distribution (Karimireddy et al., 2020; Li et al., 2020; Wang et al., 2021), a proxy dataset (Park et al., 2023), or model similarity (Blanchard et al., 2017; Sun et al., 2019; Ye et al., 2023b) yield uncertainty values primarily within the 0.75 to 2.0 range. This suggests that these approaches classify samples with lower confidence. The trends presented in Figure 2a and 2b also provide intuitive visual support for the results detailed in Table 1 and 2.

Table 2: Comparison of UFL with U-Agg and advanced aggregation strategies (%).

| METHOD | CIFAR-10 | MNIST | FASHION-MNIST | SVHN | CIFAR-100 | TINYIMAGENET-200 |
|---|---|---|---|---|---|---|
| FEDAVG | 86.02 (↑2.52) | 99.03 (↑0.15) | 91.82 (↑0.75) | 92.95 (↑2.04) | 62.44 (↑4.12) | 48.16 (↑3.86) |
| KRUM | 85.00 (↑3.54) | 99.02 (↑0.16) | 91.62 (↑0.95) | 90.40 (↑4.59) | 61.69 (↑4.87) | 47.40 (↑4.62) |
| NORM BOUND | 85.47 (↑3.07) | 98.94 (↑0.24) | 91.68 (↑0.89) | 92.77 (↑2.22) | 61.18 (↑5.38) | 48.39 (↑3.63) |
| FEDDISCO | 85.54 (↑3.00) | 99.13 (↑0.05) | 91.95 (↑0.62) | 93.91 (↑1.08) | 63.07 (↑3.49) | 48.88 (↑3.14) |
| UFL (+U-AGG) | **88.54** | **99.18** | **92.57** | **94.99** | **66.56** | **52.02** |

## 5.3 AGGREGATION EXPERIMENTS

In this experiment, Krum (Blanchard et al., 2017) and Norm Bound (Sun et al., 2019) represent classic parameter-based aggregation methods, while FedDisco (Ye et al., 2023b) leverages local datasets' statistics for enhancing training and aggregation. From Table 2, it is evident that UFL not only achieves advantages under the traditional average aggregation method (Ours+Average) but also further enhances global model performance with its proposed U-Agg aggregation method (Ours+U-Agg). On the CIFAR-10 dataset, UFL (Ours+U-Agg) achieves 88.54% accuracy, surpassing FedAvg by 2.52% and FedDisco by 2.98%. Similarly, MNIST and Fashion-MNIST, UFL performs exceptionally well with accuracies of 99.18% and 92.57%. For CIFAR-100, UFL (Ours+U-Agg) achieves 66.56% accuracy, outperforming FedDisco's 63.07% by 3.49%. On TinyImageNet-200, UFL reaches 52.02%, a 3.14% improvement over FedDisco. Compared to parameter-based aggregation methods, Krum and Norm Bound, UFL is better suited to handle variations in client data distributions. Additionally, unlike FedDisco, which relies on local data statistics, UFL incorporates sample difficulty information to optimize the aggregation strategy further, showcasing the greater potential for improvements in complex tasks.

## 5.4 ABLATION EXPERIMENTS

**High-uncertainty sample Ratio-$|\mathcal{I}|/|\mathcal{D}_k|$:** As shown in Table 3, even selecting the top 5% of local datasets $\mathcal{D}_k$ as high-uncertainty samples by uncertainty can effectively enhance $\theta^t$ performance. This selection of 5% high-uncertainty samples neither requires constructing proxy datasets nor extracting statistical characteristics, avoiding additional communication or computation overhead. Compared to FedAvg, FedDefender, and FedDisco, UFL achieves 66.04% accuracy on CIFAR-100 by simply adjusting the weight of 5% of local data samples in the loss. On TinyImageNet-200, UFL with

$|\mathcal{I}|/|\mathcal{D}_k| = 5\%$ achieves 51.80% accuracy, outperforming FedAvg (48.16%), FedDefender (49.84%), and FedDisco (48.88%). Similarly, in tasks with $C = 10$, UFL demonstrates varying degrees of improvement over other methods. Selecting the hardest 5% of local samples for loss adjustment mitigates DH and balances cost with performance, establishing a key threshold.

Table 3: The impact of $\mathcal{I}$, $N$-Passes and $\alpha$ on global model performance (%).

| $\mathcal{I}$ OF $|\mathcal{D}_k|$ | CIFAR-10 | MNIST | FASHION-MNIST | SVHN | CIFAR-100 | TINYIMAGENET-200 |
|---|---|---|---|---|---|---|
| FEDAVG | 86.02 | 99.03 | 91.82 | 92.95 | 62.44 | 48.16 |
| FEDDISCO | 85.54 | 99.13 | 91.95 | 93.91 | 63.07 | 48.88 |
| FEDDEFENDER | 87.95 | 99.02 | 92.49 | 94.55 | 65.44 | 49.84 |
| UFL ($|\mathcal{I}|/|\mathcal{D}_k| = 5\%$) | **88.48** | **99.18** | **92.56** | **94.91** | **66.04** | **51.80** |
| UFL ($|\mathcal{I}|/|\mathcal{D}_k| = 10\%$) | 88.51 (↓0.03) | 99.17 (↑0.01) | 92.39 (↑0.17) | 94.86 (↑0.05) | 66.12 (↓0.08) | 51.36 (↑0.44) |
| UFL ($|\mathcal{I}|/|\mathcal{D}_k| = 20\%$) | 88.34 (↑0.14) | 99.18 (↑0.00) | 92.50 (↑0.06) | 94.91 (↑0.00) | 66.94 (↓0.90) | 51.67 (↑0.13) |
| UFL ($|\mathcal{I}|/|\mathcal{D}_k| = 30\%$) | 88.54 (↓0.06) | 99.18 (↑0.00) | 92.57 (↓0.01) | 94.99 (↓0.08) | 66.56 (↓0.52) | 52.02 (↓0.22) |
| UFL ($|\mathcal{I}|/|\mathcal{D}_k| = 40\%$) | 88.58 (↓0.10) | 99.14 (↑0.04) | 92.62 (↓0.06) | 95.02 (↓0.11) | 66.65 (↓0.61) | 52.24 (↓0.44) |
| UFL ($|\mathcal{I}|/|\mathcal{D}_k| = 50\%$) | 88.44 (↑0.04) | 99.15 (↑0.03) | 92.42 (↑0.14) | 94.87 (↑0.04) | 66.10 (↓0.06) | 52.40 (↓0.60) |
| UFL ($\alpha = 0.2$) | **88.54** | **99.18** | **92.57** | **94.99** | **66.56** | **52.02** |
| UFL ($\alpha = 0.4$) | 88.43 (↑0.11) | 99.21 (↓0.03) | 92.71 (↓0.14) | 95.02 (↓0.03) | 65.64 (↑0.92) | 50.46 (↑1.56) |
| UFL ($\alpha = 0.6$) | 88.2 (↑0.34) | 99.18 (↓0.0) | 92.52 (↑0.05) | 94.8 (↑0.19) | 65.3 (↑1.26) | 49.38 (↑2.64) |
| UFL ($\alpha = 0.8$) | 88.17 (↑0.37) | 99.16 (↑0.02) | 92.66 (↓0.09) | 94.94 (↑0.05) | 64.91 (↑1.65) | 47.96 (↑4.06) |
| UFL ($\alpha = 1.0$) | 88.19 (↑0.35) | 99.19 (↓0.01) | 92.6 (↓0.03) | 95.03 (↓0.04) | 63.59 (↑2.97) | 47.06 (↑4.96) |
| UFL (N-PASSES=5) | **88.50** | **99.22** | **92.53** | **94.87** | **66.46** | **51.56** |
| UFL (N-PASSES=10) | 88.54 (↓0.04) | 99.18 (↑0.04) | 92.57 (↓0.04) | 94.99 (↓0.12) | 66.56 (↓0.10) | 52.02 (↓0.46) |
| UFL (N-PASSES=15) | 88.68 (↓0.18) | 99.16 (↑0.06) | 92.64 (↓0.11) | 95.00 (↓0.13) | 66.79 (↓0.33) | 51.64 (↓0.08) |

$N$**-Forward Passes of $\theta^t$ with MC Dropout**: We set $N$-passes to 5, 10, and 15 to explore how the number of predictions affects the performance of the global model. In Table 3, it is evident that the UFL method enhances global model performance with just 5 forward passes. On CIFAR-10, UFL ($N$-passes=5) achieves an accuracy of 88.50%, far surpassing FedAvg (86.02%) and FedDisco (85.54%). Similarly, UFL performs exceptionally well on MNIST, Fashion-MNIST, and SVHN. On more challenging datasets like CIFAR-100 and TinyImageNet-200, UFL achieves accuracies of 66.46% and 51.56%, with 5 $N$-passes. Performance gains are limited when $N$-passes increase to 10 or 15, making 5 $N$-passes a key balance point between performance and computational efficiency.

**Weight Scaling Factor of high-uncertainty sample Loss-$\alpha$**: Table 3 clearly shows that as $\alpha$ gradually increases to 1.0, the accuracy of UFL on the CIFAR-10 dataset decreases from 66.56% to 63.59%. This trend is even more pronounced on more challenging datasets such as CIFAR-100 and TinyImageNet-200, where accuracy drops from 52.02% to 47.06%. In contrast, UFL demonstrates higher stability on datasets like MNIST, SVHN, and Fashion-MNIST, with accuracy remaining nearly unaffected by changes in $\alpha$, maintaining high levels of 99.19% and 95.03%, even when $\alpha$ reaches 1.0. To consistently outperform existing methods such as FedAvg, FedDisco, and FedDefender across diverse datasets, $\alpha$ should be controlled within the range of 0.2 to 0.4 to balance performance.

# 6 CONCLUSION

This paper addresses challenges in FL beyond distributional heterogeneity, uncovering a pronounced long-tail effect in sample uncertainty—phenomena that appear intrinsic to the FL paradigm rather than mere consequences of data skew. To tackle these sample-level uncertainties, we propose Uncertainty-driven Federated Learning (UFL). UFL employs MC dropout to quantify sample uncertainty, adaptively re-weights local loss functions to emphasize high-uncertainty instances, and thereby mitigates the uncertainty long tail while improving feature representation. At the server side, UFL introduces U-Agg, an uncertainty-aware aggregation strategy that limits the propagation of high-uncertainty samples. Theoretical analysis guarantees the convergence of UFL and the correctness of U-Agg. Extensive experiments demonstrate that UFL consistently outperforms eight state-of-the-art FL methods across datasets such as CIFAR-100 and TinyImageNet-200. These gains stem from UFL's ability to directly address the long-tail distribution of sample uncertainty and improve the representation of high-uncertainty samples, validating our core hypothesis. By shifting attention from coarse distributional statistics to fine-grained sample-level uncertainty, UFL may offer a new and complementary perspective for understanding and mitigating performance degradation in FL.

ETHICS STATEMENT

This work uses only computational methods and publicly available datasets, with no human subjects or private data. It follows the ICLR Code of Ethics, with no conflicts of interest. While acknowledging potential dual-use concerns, we stress responsible deployment and adhere to research integrity. All methods and results are reported transparently to support reproducibility.

REPRODUCIBILITY STATEMENT

We provide implementation details in the Appendix and source code at the anonymous link https://anonymous.4open.science/r/UFL-B0FC to support reproduction of the main results.

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

## A    STATEMENT OF THE USE OF LARGE LANGUAGE MODELS (LLMs)

In this paper, we just used the LLM, ChatGPT, to polish the language of the paper. We did not use LLMs to generate any content or ideas in this work. We have verified the accuracy of all content and ideas in the paper.

## B    UFL FRAMEWORK

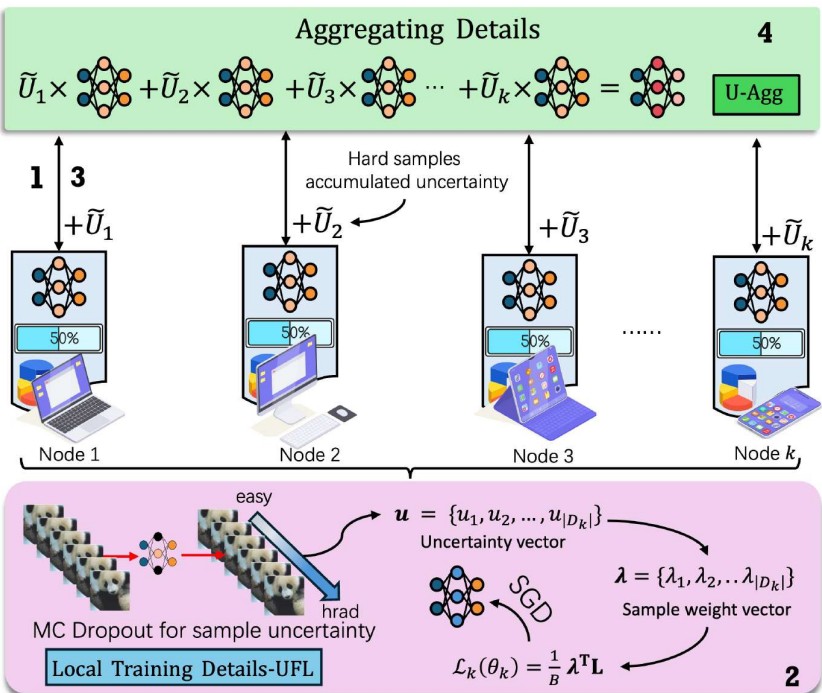

Figure 3: The pipeline of proposed UFL and U-Agg. 1. Local users download the global model $\theta^t$. 2. Each node uses $\theta^t$ to perform MC dropout for uncertainty estimation, then trains $\mathcal{D}_k$ based on $\theta^t$ to obtain the local model $\theta_k^t$. 3. Nodes then upload $\theta_k^t$ and $\widetilde{U}_k$. 4. The server aggregates all local models based on $\widetilde{U}_k$.

## C    THE PROCEDURE OF ALGORITHMS

---

**Algorithm 1** `UFL: Uncertainty-driven Federated Learning`

---

1: **Input:**
2:     $K$: Number of clients
3:     $\mathcal{D}_k$: Local dataset for client $k$, where $|\mathcal{D}_k|$ is the dataset size
4:     $T$: Number of global training rounds
5:     $\alpha$: Scaling factor for high-uncertainty sample weights
6:     $\mathcal{I}$: Number of high-uncertainty samples per client (e.g., 5% of $|\mathcal{D}_k|$)
7:     $B$: Batch size for local training
8:     $N$: Number of Monte Carlo Dropout passes
9: **Initialization:**
10:     Initialize global model parameters $\theta^0$, Set learning rate $\eta$, and optimizer (e.g., SGD)
11: **for** round $t = 0, 1, \ldots, T-1$ **do**                                    ▷ Start FL local training
12:     Server sends $\theta^t$ to all $K$ clients
13:     Client updates local model: $\theta_k^t \leftarrow \theta^t$
14:     **for** each client $k = 1, 2, \ldots, K$ in parallel **do**
15:         Initialize uncertainty vector $\boldsymbol{u} \leftarrow [\,]$
16:         **for** each sample $(x_i, y_i) \in \mathcal{D}_k$ **do**
17:             Compute average prediction $\bar{p}_{i,c}$ for class $c$ using $\theta^t$ and $N$ passes
18:             Compute uncertainty: $u_i = -\sum_{c=1}^{C} y_{i,c} \log(\bar{p}_{i,c})$
19:             Append $u_i$ to $\boldsymbol{u}$
20:         **end for**
21:         Sort $\boldsymbol{u}$ in descending order to get indices of top $\mathcal{I}$% samples
22:         **if** $i$ in top $|\mathcal{I}|$ indices **then**
23:             $\lambda_i \leftarrow 1 + \alpha \cdot u_i$                          ▷ For high-uncertainty samples
24:         **else**
25:             $\lambda_i \leftarrow 1$                                              ▷ For easy samples
26:         **end if**
27:         Initialize local model $\theta_k^{t+1} = \theta^t$
28:         **for** each batch of $B$ samples $\{(x_i, y_i)\}_{i=1}^{B} \subset \mathcal{D}_k$ **do**
29:             Compute per-sample loss: $l(x_i, y_i; \theta_k^t) = -\lambda_i \sum_{c=1}^{C} y_{i,c} \log(p_{i,c})$
30:             Compute batch loss: $F_k(\theta_k^t) = \frac{1}{B} \sum_{i=1}^{B} l(x_i, y_i; \theta_k^t)$
31:             Update $\theta_k^{t+1} \leftarrow \theta_k^t - \eta \nabla F_k(\theta_k^t)$
32:         **end for**
33:         Compute $U_k = \sum_{i=1}^{|\mathcal{I}|} u_i$ for top $|\mathcal{I}|$ high-uncertainty samples
34:         Upload $\theta_k^{t+1}$ and $U_k$ to server
35:     **end for**
36:     **Server Aggregation:**                                                       ▷ U-Agg
37:         Collect $\{\theta_k^{t+1}, U_k\}_{k=1}^{K}$ from all clients
38:         Normalize uncertainties: $\hat{U}_k = \frac{U_k}{\sum_{i=1}^{K} U_i}$ for $k = 1, \ldots, K$
39:         Compute relative weights: $w_k = 1 - \hat{U}_k$
40:         Normalize weights: $\widetilde{U}_k = \frac{w_k}{\sum_{i=1}^{K} w_i}$
41:         Update global model: $\theta^{t+1} = \sum_{k=1}^{K} \widetilde{U}_k \theta_k^{t+1}$
42: **end for**
43: **Output:** Final global model $\theta^T$

---

## D    FEDERATED LEARNING

FL (McMahan et al., 2017) relies on collaborative training and the aggregation of model parameters from multiple parties. Suppose there are $K$ nodes, each with a local dataset $\mathcal{D}_k = (x, y)$, where $x$ represents the samples and $y$ represents the labels. Let the local loss $F_k(\theta_k)$ for the local model $\theta_k$ be defined as follows:

$$F_k(\theta_k) = \frac{1}{|\mathcal{D}_k|} \sum l(x_i, y_i; \theta_k), \ (x_i, y_i) \in \mathcal{D}_k. \tag{11}$$

$l\left(x_i, y_i; \theta_k\right)$ denotes the loss for each sample, and $|\mathcal{D}_k|$ represents the size of the local dataset. By minimizing $F_k\left(\theta_k\right)$, node $k$ can optimize its local model to fit local datasets better. The steps of FL are as follows: (1). The central server initializes the global model parameters $\theta^0$. (2). At round $t$, the server sends the global model parameters $\theta^t$ to all nodes ($\theta_k^t \leftarrow \theta^t$). Each node $k$ performs gradient descent steps on its local data $\mathcal{D}_k$:

$$\theta_k^{t+1} = \theta_k^t - \eta \nabla F_k\left(\theta_k^t\right), \tag{12}$$

where $\eta$ is the learning rate and $\nabla F_k\left(\theta_k^t\right)$ is the gradient of the local loss. The updated model parameters $\theta_k^{t+1}$ are uploaded from all nodes to the central server. (3). The server aggregates these parameters using a weighted average:

$$\theta^{t+1} = \sum_{k=1}^{K} \frac{|\mathcal{D}_k|}{|\mathcal{D}|} \theta_k^{t+1}, \tag{13}$$

where $|\mathcal{D}| = \sum_{k=1}^{K} |\mathcal{D}_k|$ is the total number of samples across all local datasets. This method, known as FedAvg (McMahan et al., 2017), updates the global model by computing a weighted average of the parameters by considering the local dataset scale. Finally, the above steps are repeated until the communication round ends.

# E   EMPIRICAL OBSERVATIONS

This section presents empirical evidence of two challenges in FL that extend beyond typical DH effects. Using the CIFAR-10 dataset (partitioned into IID and Non-IID for FL), we estimate sample uncertainty with MC Dropout and analyze model features. Our findings are twofold: (1) FL displays a characteristic long-tail effect in sample uncertainty distributions when compared to centralized learning. (2) high-uncertainty samples exhibit poor feature separability, a difficulty that persists even in IID settings. These observations highlight inherent challenges within the FL paradigm, with their detrimental effects appearing to propagate through model aggregation and impact the global model.

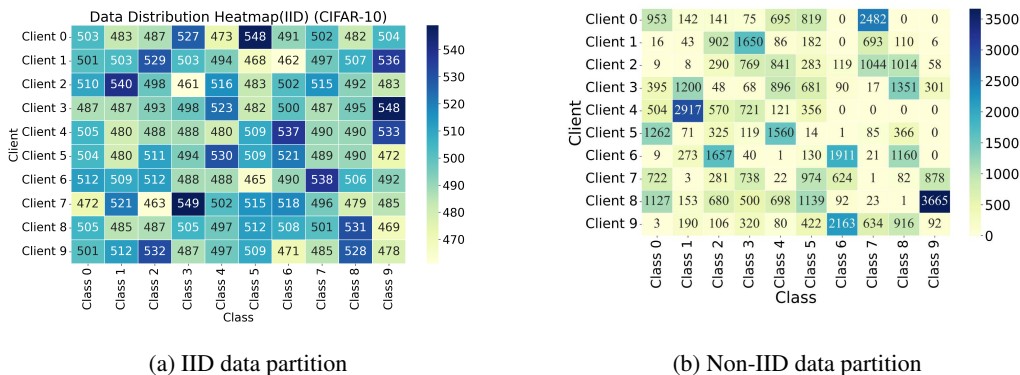

(a) IID data partition                                      (b) Non-IID data partition

Figure 4: Data Distributions: IID and Non-IID

## E.1   THE LONG-TAIL EFFECT

As illustrated in Figure 1, the distribution of sample uncertainty within the FL displays a pronounced long-tail effect. In centralized learning, the vast majority of samples achieve low uncertainty in later training stages, clustering at the head of the distribution. In contrast, FL exhibits a distribution where a comparatively larger fraction of samples retains higher uncertainty, resulting in the characteristic long tail. This is evident as a subset of samples in FL consistently presents substantially higher uncertainty than those in centralized training at similar stages. As depicted in Figure 1 (a) and (b), this phenomenon is observable across both IID and Non-IID data settings. This suggests that the FL itself may cause the difficulty models face in learning certain samples with high confidence. Notably, these high-uncertainty samples in the 'long-tail' region often become a critical bottleneck in the learning process. This effect may stem from two aspects of FL: the model aggregation process potentially averaging out or obscuring client-specific learning on local high-uncertainty samples, and the inherent challenge for a single global model to sufficiently capture and adapt to the nuanced, critical features of all samples across diverse client datasets. (See Appendix 6 for more observations.)

## E.2   FEATURE INSEPARABILITY OF HIGH-UNCERTAINTY SAMPLES

Our second key observation, illustrated by the t-SNE visualizations of intermediate layer features in Figure 1 (c), (d), and (e), is that samples identified as 'hard' (typically those with high uncertainty or prone to

misclassification) exhibit poor separability in their learned feature representations. This challenge of feature entanglement for difficult samples persists across different data distributions and model types within the FL framework. Specifically, Figure 1 (c) presents the feature space for client $D_0$'s data as processed by the $\theta^t$ under an IID setting. Even in this idealized scenario without distributional skew, the features of high-uncertainty samples appear intermingled and lack the clear clustering often seen with easier-to-learn samples. This difficulty in separating high-uncertainty sample features remains pronounced when the global model $\theta^t$ is applied to the local dataset $D_0$ of client 0 in a Non-IID setting, as shown in Figure 1 (d). Furthermore, Figure 1 (e) demonstrates this inseparability from the perspective of a local model: when client 0's local model $\theta_0^t$ processes data from a different client ($D_4$), the high-uncertainty samples from $D_4$ still result in poorly separated. Collectively, t-SNE plots in Figure 1 (c)-(e) suggest that the learning difficulty associated with these high-uncertainty samples is not solely a consequence of Non-IID. Instead, it points towards more fundamental, sample-intrinsic characteristics that impede the formation of clearly separable features by both local and global models in FL. This feature's inseparability poses an obstacle to achieving robust classification and generalization across clients.

### E.3 IMPLICATIONS: PROPAGATION OF SAMPLE-LEVEL CHALLENGES

The two preceding empirical findings, the long-tail distribution and the feature inseparability of high-uncertainty samples, collectively indicate a consequence within the FL process: the propagation of sample-level learning challenges. In FL, local models are trained on diverse datasets and are thus influenced by the specific high-uncertainty samples within each client's data. These models then contribute their learned parameters, which encapsulate any unresolved difficulties with high-uncertainty samples, to the central server for aggregation. When these local updates are combined, the learning deficiencies from various clients, particularly the challenges posed by high-uncertainty samples, can be assimilated and averaged into the updated global model. The persistence of a pronounced long-tail in sample uncertainty (Figure 1 (a), (b)) and the continued feature entanglement for certain samples in the global model (Figure 1 (c), (d)) serve as evidence for this propagation. This suggests a mechanism where difficulties initially encountered at the local level are not confined locally but are instead disseminated system-wide through the FL aggregation. Such propagation is intrinsically able to hinder the global model's capacity for effective generalization across the distributed local data. ()

## E.4 Feature Observations by Different Models on Different Local Datasets, Long Tail Effects

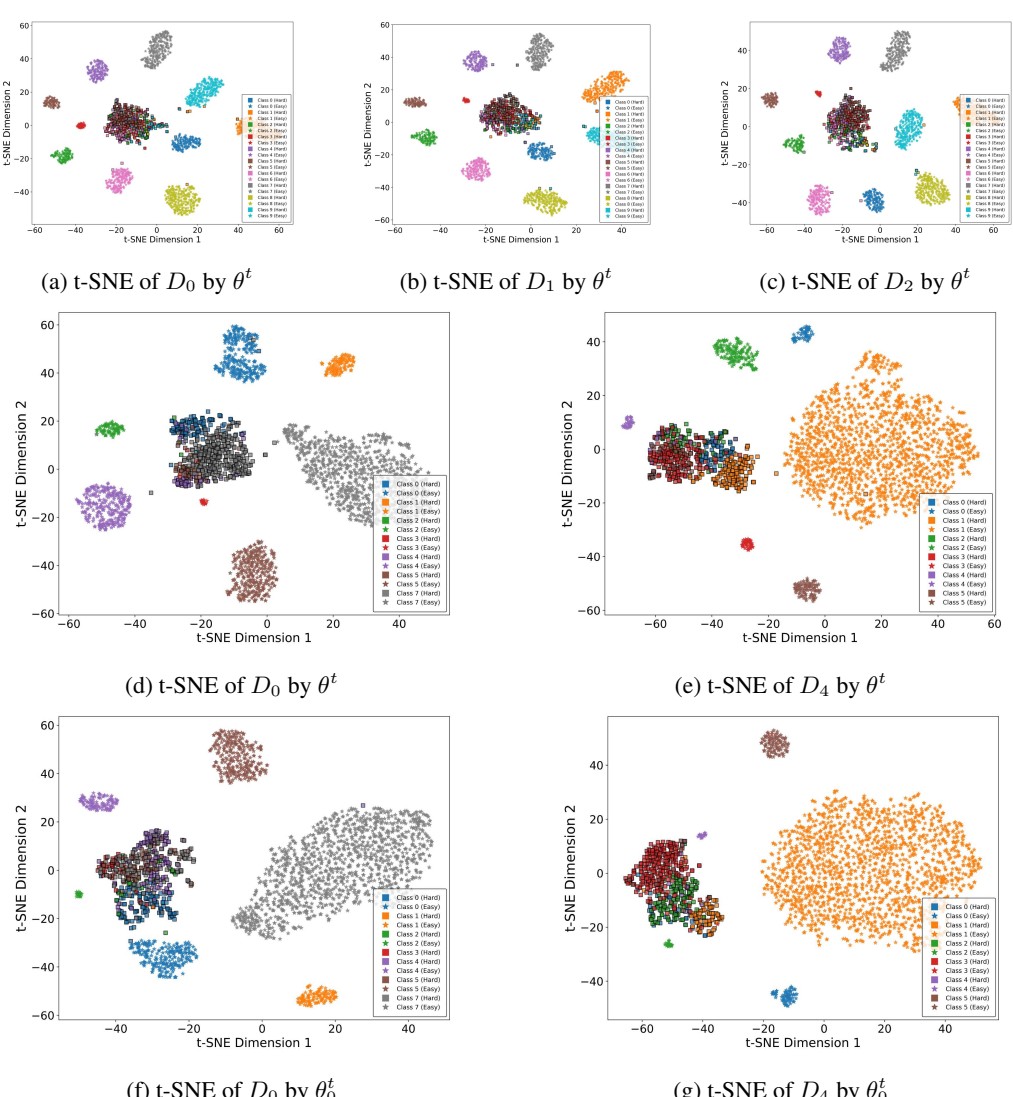

(a) t-SNE of $D_0$ by $\theta^t$      (b) t-SNE of $D_1$ by $\theta^t$      (c) t-SNE of $D_2$ by $\theta^t$

(d) t-SNE of $D_0$ by $\theta^t$             (e) t-SNE of $D_4$ by $\theta^t$

(f) t-SNE of $D_0$ by $\theta_0^t$             (g) t-SNE of $D_4$ by $\theta_0^t$

Figure 5: Empirical observations of DH. $\theta_k^t$ is the local model. (a) - (c) present the t-SNE visualization of intermediate layer features for the global model $\theta^t$ with different local datasets under independent and identically distributed (IID). (d) - (g) present the t-SNE visualization of intermediate layer features for global and local models handling different datasets under Non-IID. The $\mathcal{D}_k$ is the local dataset for client $k$, and $\theta^t$ denotes the global model.

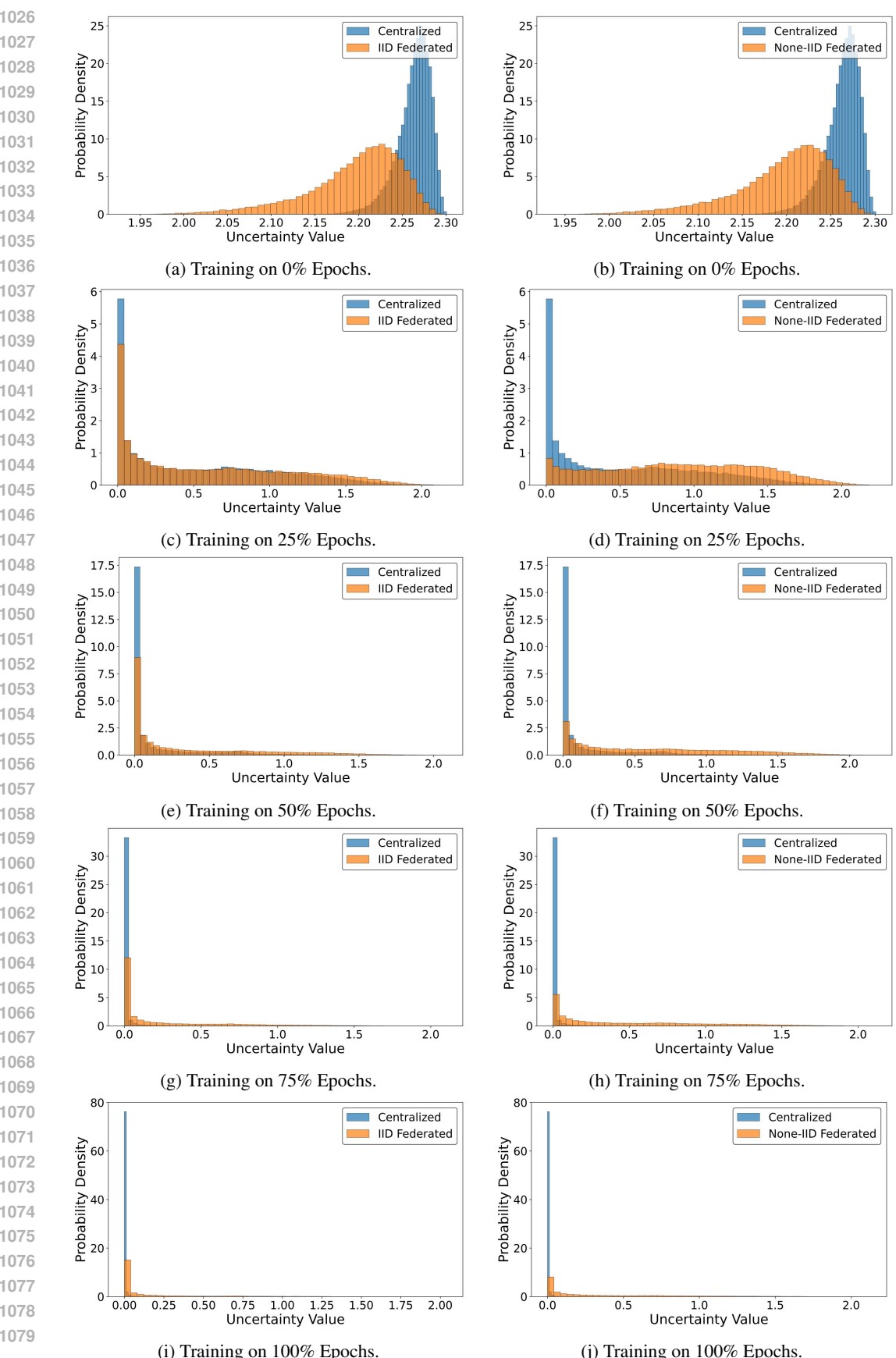

Figure 6: (a)–(e) illustrate the long tail effect when comparing FL with centralized learning under an IID data distribution, while (f)–(j) illustrate the long tail effect under a Non-IID data distribution.

## F  RATIONALE OF ASSUMPTION 4.4

**Uncertainty-Modulated Gradient Divergence**  The expected squared divergence between a client's local gradient and the global average gradient $\nabla F(\theta) = \sum_{j=1}^{K} \frac{|D_j|}{|\mathcal{D}|} \nabla F_j(\theta_j)$ is bounded, and this bound is influenced by the client's relative uncertainty. Specifically, there exist non-negative constants $B_G^2$ and $V$ such that for all clients $k$ and parameters $\theta$:

$$\mathbb{E}||\nabla F_k(\theta_k) - \nabla F(\theta)||^2 \leq B_G^2 + V\hat{U}_k \tag{14}$$

*Rationale:* This assumption connects higher client uncertainty $\hat{U}_k$ to a potentially larger deviation of its local gradient $\nabla F_k$ from the global average gradient $\nabla F$. This aligns with intuition for two main reasons. First, clients whose local data characteristics differ from the global average might naturally have different gradient directions. Second, high uncertainty $\hat{U}_k$ itself indicates a poor fit of the current model for client $k$'s data. In either scenario, these clients are expected to contribute gradients less aligned with the overall global objective, thus exhibiting greater divergence $||\nabla F_k - \nabla F||^2$. $B_G^2$ represents the baseline heterogeneity caused by data diversity, while the $V\hat{U}_k$ term models the additional divergence attributed to the measured uncertainty.

## G  DETAILED PROOF OF THEOREM 4.5

This appendix provides a more detailed proof for Theorem 4.5.

*Proof:* The proof follows standard FL convergence analysis, e.g., bounding $\mathbb{E}[F(\theta^{t+1})] - F(\theta^t)$ using the descent lemma. Key steps involve bounding the inner product $\langle \nabla F(\theta^t), \theta^{t+1} - \theta^t \rangle$ and the squared update norm $||\theta^{t+1} - \theta^t||^2$. The analysis needs to account for $\tau$ local steps and the variance from stochastic gradients (Assumption 4.3). Crucially, terms related to client drift or the difference between the aggregated update and the true global gradient $\nabla F(\theta^t)$ are bounded using Assumption 4.4. This introduces the dependence on $\sum p_k(B_G^2 + V\hat{U}_k)$. Summing over $T$ rounds and rearranging yields the bound.

We analyze the convergence of a generalized FedAvg algorithm where $K$ clients are selected (or all participate, $p_k$ are weights, $\sum p_k = 1$), each performing $\tau$ local SGD steps with learning rate $\eta$ starting from $\theta^t$. Let $\theta_{k,s}^t$ denote the model parameters on client $k$ after $s$ local steps in round $t$, so $\theta_{k,0}^t = \theta^t$. The local update is $\theta_{k,s+1}^t = \theta_{k,s}^t - \eta g_k(\theta_{k,s}^t; \xi_{k,s})$, where $g_k(\cdot; \xi_{k,s})$ is the stochastic gradient on minibatch $\xi_{k,s}$. The effective update from client $k$ is $\Delta_k^t = \theta^t - \theta_{k,\tau}^t = \eta \sum_{s=0}^{\tau-1} g_k(\theta_{k,s}^t; \xi_{k,s})$. The global update is $\theta^{t+1} = \theta^t - \sum_{k=1}^{K} p_k \Delta_k^t = \theta^t - \eta \sum_{k=1}^{K} p_k \sum_{s=0}^{\tau-1} g_k(\theta_{k,s}^t; \xi_{k,s})$. For simplicity in analysis, we often use an effective server learning rate $\eta_{eff}$ and write $\theta^{t+1} = \theta^t - \eta_{eff}\bar{g}^t$, where $\bar{g}^t = \frac{1}{\eta_{eff}} \sum p_k \Delta_k^t$. If $\eta_{eff} = 1$, $\bar{g}^t = \sum p_k \Delta_k^t$. Let's assume $\eta_{eff} = 1$ for model averaging, so $\theta^{t+1} = \sum p_k \theta_{k,\tau}^t$.

By the $L$-smoothness of $F(\theta)$ (Assumption 4.1), the descent lemma holds:

$$\mathbb{E}[F(\theta^{t+1})] \leq \mathbb{E}[F(\theta^t)] + \mathbb{E}[\langle \nabla F(\theta^t), \theta^{t+1} - \theta^t \rangle] + \frac{L}{2}\mathbb{E}[||\theta^{t+1} - \theta^t||^2] \tag{15}$$

where $\mathbb{E}[\cdot]$ is the expectation over all randomness up to round $t$.

We need to bound the two expectation terms. Let $\mathbb{E}_t[\cdot]$ denote expectation conditional on $\theta^t$.

**1. Bounding the Inner Product Term** $\mathbb{E}_t[\langle \nabla F(\theta^t), \theta^{t+1} - \theta^t \rangle]$**:**

$$\mathbb{E}_t[\langle \nabla F(\theta^t), \theta^{t+1} - \theta^t \rangle] = \mathbb{E}_t[\langle \nabla F(\theta^t), -\eta \sum_{k=1}^{K} p_k \sum_{s=0}^{\tau-1} g_k(\theta_{k,s}^t; \xi_{k,s}) \rangle]$$

$$= \underbrace{-\eta \sum_{k=1}^{K} p_k \sum_{s=0}^{\tau-1} \mathbb{E}_t[\langle \nabla F(\theta^t), \mathbb{E}_{\xi_{k,s}}[g_k(\theta_{k,s}^t; \xi_{k,s})] \rangle]}_{\text{(Linearity of expectation)}} = \underbrace{-\eta \sum_{k=1}^{K} p_k \sum_{s=0}^{\tau-1} \mathbb{E}_t[\langle \nabla F(\theta^t), \nabla F_k(\theta_{k,s}^t) \rangle]}_{\text{(Assumption 4.3: Unbiased gradients)}}$$

$$= \underbrace{-\eta \sum_{k=1}^{K} p_k \sum_{s=0}^{\tau-1} \mathbb{E}_t \left[ \frac{1}{2}||\nabla F(\theta^t)||^2 + \frac{1}{2}||\nabla F_k(\theta_{k,s}^t)||^2 - \frac{1}{2}||\nabla F(\theta^t) - \nabla F_k(\theta_{k,s}^t)||^2 \right]}_{2<a,b>=||a||^2+||b||^2-||a-b||^2}$$

$$\leq \underbrace{-\eta\tau||\nabla F(\theta^t)||^2 + \eta^2\tau L^2 C_{drift} + \eta \sum_{k=1}^{K} p_k \sum_{s=0}^{\tau-1} \mathbb{E}_t||\nabla F_k(\theta^t) - \nabla F_k(\theta_{k,s}^t)||^2}_{\text{Standard bounds using L-smoothness (details omitted, see Scaffold (Karimireddy et al., 2020) for details)}}$$

$$\leq \underbrace{-\eta\tau||\nabla F(\theta^t)||^2 + \eta^2\tau L^2 C_{drift} + \eta L^2 \sum_{k=1}^{K} p_k \sum_{s=0}^{\tau-1} \mathbb{E}_t||\theta^t - \theta_{k,s}^t||^2}_{\text{Simplified structure, } C_{drift} \text{ depends on } \sigma^2, \tau)}$$

$$\leq \underbrace{-\eta\tau||\nabla F(\theta^t)||^2 + \eta^2\tau L^2 C_{drift} + \eta^3 L^2 \tau^3 (\text{Const})(\mathbb{E}_t[\sum p_k ||\nabla F_k(\theta^t)||^2] + \sigma^2)}_{\text{Bound } ||\theta^t - \theta_{k,s}^t||^2 <= s^2 * \eta^2 * \text{(gradient norms + variance)} \approx s^2 * \eta^2 * (||\nabla F_k||^2 + \sigma^2)}$$

$$\leq -\frac{\eta\tau}{2}||\nabla F(\theta^t)||^2 + \frac{\eta\tau}{2}\mathbb{E}_t||\nabla F(\theta^t) - \sum_k p_k \nabla F_k(\theta^t)||^2 + C_1'\eta^2\tau(\tau-1)\sigma^2 + C_2'\eta^3\tau^2 L^2\sigma^2$$

$$(16)$$

The second term $\frac{\eta\tau}{2}\mathbb{E}_t||\nabla F(\theta^t) - \sum_k p_k \nabla F_k(\theta^t)||^2$ captures the initial gradient dissimilarity. The $\sigma^2$ terms arise from the variance accumulated over $\tau$ steps.

**2. Bounding the Squared Norm Term** $\mathbb{E}_t[||\theta^{t+1} - \theta^t||^2]$**:**

$$\mathbb{E}_t[||\theta^{t+1} - \theta^t||^2] = \mathbb{E}_t[||\eta \sum_{k=1}^{K} p_k \sum_{s=0}^{\tau-1} g_k(\theta_{k,s}^t; \xi_{k,s})||^2]$$

$$= \underbrace{\eta^2 ||\mathbb{E}_t[\sum_{k=1}^{K} p_k \sum_{s=0}^{\tau-1} g_k(\theta_{k,s}^t; \xi_{k,s})]||^2 + \eta^2 \mathbb{E}_t[||\sum_{k=1}^{K} p_k \sum_{s=0}^{\tau-1}(g_k(\theta_{k,s}^t; \xi_{k,s}) - \mathbb{E}_\xi[g_k])||^2]}_{E[||X||^2]=||E[X]||^2+\text{Var}(X)}$$

$$\approx \eta^2 ||\sum_{k=1}^{K} p_k \sum_{s=0}^{\tau-1} \nabla F_k(\theta_{k,s}^t)||^2 + (\text{Variance term})$$

$$\leq \underbrace{2\eta^2\tau^2 ||\sum_{k=1}^{K} p_k \nabla F_k(\theta^t)||^2 + 2\eta^2 \sum_{k,s} p_k^2 \mathbb{E}_t||\nabla F_k(\theta_{k,s}^t) - \nabla F_k(\theta^t)||^2 + (\text{Variance term})}_{\text{Bounding sum over } \tau \text{ steps using } ||a+b||^2 \leq 2||a||^2+2||b||^2 \text{ \& Jensen/Cauchy}}$$

$$\leq 2\eta^2\tau^2 ||\nabla F(\theta^t)||^2 + 2\eta^2\tau^2 ||\sum p_k(\nabla F_k(\theta^t) - \nabla F(\theta^t))||^2 + C_1''\eta^4\tau^4 L^2\sigma^2 + (\text{Variance term})$$

$$+ \eta^2\tau\sigma^2 \sum_{k=1}^{K} p_k^2 \quad (\text{Simplified variance bound, assumes independence across clients/steps})$$

$$\leq \underbrace{C_2''\eta^2\tau^2 ||\nabla F(\theta^t)||^2 + C_3''\eta^2\tau^2 \mathbb{E}_t||\sum p_k(\nabla F_k(\theta^t) - \nabla F(\theta^t))||^2 + C_4''\eta^2\tau\sigma^2 \sum \frac{p_k^2}{|\mathcal{D}_k|/|\mathcal{D}|}}_{\text{Apply standard inequalities (Jensen, Cauchy-Schwarz) \& bounds on SGD variance/drift}}$$

$$(17)$$

The factor $\sum p_k^2/(|\mathcal{D}_k|/|\mathcal{D}|)$ in the variance term is heuristic here; precise bounds depend on sampling assumptions.

**3. Bounding Gradient Divergence Term (Using Assumption 4.4):** We need to bound $\mathbb{E}_t||\sum_{k=1}^{K} p_k(\nabla F_k(\theta^t) - \nabla F(\theta^t))||^2$.

$$\mathbb{E}_t||\sum_{k=1}^{K} p_k(\nabla F_k(\theta^t) - \nabla F(\theta^t))||^2 \leq \mathbb{E}_t \sum_{k=1}^{K} p_k ||\nabla F_k(\theta^t) - \nabla F(\theta^t)||^2 \quad \text{(Jensen's inequality)}$$

$$\leq \sum_{k=1}^{K} p_k (B_G^2 + V\hat{U}_k) \quad \text{(Using Assumption 4.4)}$$

$$= B_G^2 + V \sum_{k=1}^{K} p_k \hat{U}_k \tag{18}$$

Also note that $\mathbb{E}_t||\nabla F(\theta^t) - \sum_k p_k \nabla F_k(\theta^t)||^2 = \mathbb{E}_t||\sum_k p_k(\nabla F(\theta^t) - \nabla F_k(\theta^t))||^2 \leq B_G^2 + V\sum p_k \hat{U}_k$.

**4. Combining and Rearranging:** Substitute bounds equation 16, equation 17, and equation 18 into the descent lemma equation 15. Let $\eta_{eff} = \eta$.

$$\mathbb{E}_t[F(\theta^{t+1})] \leq F(\theta^t) - \frac{\eta\tau}{2}||\nabla F(\theta^t)||^2 + \frac{\eta\tau}{2}(B_G^2 + V\sum p_k \hat{U}_k) + C_1'\eta^2\tau(\tau-1)\sigma^2 + C_2'\eta^3\tau^2 L^2\sigma^2$$

$$+ \frac{L}{2}\left[C_2''\eta^2\tau^2||\nabla F(\theta^t)||^2 + C_3''\eta^2\tau^2(B_G^2 + V\sum p_k \hat{U}_k) + C_4''\eta^2\tau\sigma^2 \sum \frac{p_k^2}{|\mathcal{D}_k|/|\mathcal{D}|}\right]$$

Rearrange to isolate $||\nabla F(\theta^t)||^2$:

$$\left(\frac{\eta\tau}{2} - \frac{LC_2''\eta^2\tau^2}{2}\right)||\nabla F(\theta^t)||^2 \leq F(\theta^t) - \mathbb{E}_t[F(\theta^{t+1})]$$

$$+ \left(\frac{\eta\tau}{2} + \frac{LC_3''\eta^2\tau^2}{2}\right)(B_G^2 + V\sum p_k \hat{U}_k)$$

$$+ C_1'\eta^2\tau(\tau-1)\sigma^2$$

$$+ C_2'\eta^3\tau^2 L^2\sigma^2$$

$$+ \frac{LC_4''\eta^2\tau\sigma^2}{2} \sum \frac{p_k^2}{|\mathcal{D}_k|/|\mathcal{D}|}$$

Assume $\eta$ is chosen small enough such that $\frac{\eta\tau}{2} - \frac{LC_2''\eta^2\tau^2}{2} > 0$, let this coefficient be $C_{grad} > 0$.

$$||\nabla F(\theta^t)||^2 \leq \frac{1}{C_{grad}}(F(\theta^t) - \mathbb{E}_t[F(\theta^{t+1})])$$

$$+ \frac{1}{C_{grad}}\left(\frac{\eta\tau}{2} + \frac{LC_3''\eta^2\tau^2}{2}\right)(B_G^2 + V\sum p_k \hat{U}_k)$$

$$+ \frac{C_1'\eta^2\tau(\tau-1)\sigma^2}{C_{grad}}$$

$$+ \frac{C_2'\eta^3\tau^2 L^2\sigma^2}{C_{grad}}$$

$$+ \frac{LC_4''\eta^2\tau\sigma^2}{2C_{grad}} \sum \frac{p_k^2}{|\mathcal{D}_k|/|\mathcal{D}|}$$

**5. Summing over $T$ Rounds and Taking Total Expectation:** Take the total expectation $\mathbb{E}[\cdot]$ over all rounds. Sum from $t = 0$ to $T - 1$:

$$\sum_{t=0}^{T-1} \mathbb{E}||\nabla F(\theta^t)||^2 \leq \frac{1}{C_{grad}} \sum_{t=0}^{T-1} (\mathbb{E}[F(\theta^t)] - \mathbb{E}[F(\theta^{t+1})]) + T \times \text{(Constant error terms)}$$

$$= \frac{1}{C_{grad}}(\mathbb{E}[F(\theta^0)] - \mathbb{E}[F(\theta^T)]) + T \times \text{(Constant error terms)}$$

$$\leq \frac{F(\theta^0) - F_{\inf}}{C_{grad}} + T \times \text{(Constant error terms)} \quad \text{(Using Assumption 4.2)}$$

Divide by $T$:

$$\frac{1}{T}\sum_{t=0}^{T-1}\mathbb{E}||\nabla F(\theta^t)||^2 \leq \frac{F(\theta^0) - F_{\text{inf}}}{TC_{grad}}$$

$$+ \frac{1}{C_{grad}}\left(\frac{\eta\tau}{2} + \frac{LC_3''\eta^2\tau^2}{2}\right)(B_G^2 + V\mathbb{E}[\sum p_k\hat{U}_k])$$

$$+ \frac{C_1'\eta^2\tau(\tau-1)\sigma^2}{C_{grad}}$$

$$+ \frac{C_2'\eta^3\tau^2L^2\sigma^2}{C_{grad}}$$

$$+ \frac{LC_4''\eta^2\tau\sigma^2}{2C_{grad}}\mathbb{E}[\sum \frac{p_k^2}{|\mathcal{D}_k|/|\mathcal{D}|}]$$

This matches the structure of Theorem 4.5 (Eq. equation 10), where $C_1, \ldots, C_5$ absorb the complex coefficients derived above. For instance: $C_1 \propto 1/C_{grad}$, $C_2 \propto LC_4''\eta\tau\sigma^2/(2C_{grad})$ (related to the variance term with $p_k^2$), $C_3 \propto C_1'\eta\tau(\tau-1)\sigma^2/C_{grad}$ (related to drift variance), $C_4 \propto (\frac{\eta\tau}{2} + \frac{LC_3''\eta^2\tau^2}{2})V/C_{grad}$ (related to the uncertainty term $V\sum p_k\hat{U}_k$)

This detailed derivation shows how the terms involving $\sigma^2$, $\tau$, $B_G^2$, and $V\hat{U}_k$ arise from the analysis of local steps and gradient divergence.

## H   DERIVATION OF OPTIMAL WEIGHTS (PROPOSITION 4.6)

This appendix derives the optimal aggregation weights $p_k^*$ form stated in Proposition 4.6. We aim to minimize the dominant terms in the convergence bound from Theorem 4.5 (Eq. equation 10) that depend on the weights $p_k = \{p_1, \ldots, p_K\}$, subject to $\sum p_k = 1, p_k \geq 0$.

The relevant terms from the bound to minimize are approximately:

$$J(p) = C_2\eta_{\text{eff}}\sigma^2\sum_{k=1}^{K}\frac{p_k^2}{|\mathcal{D}_k|/|\mathcal{D}|} + C_4\eta_{\text{eff}}\tau\sum_{k=1}^{K}p_k(B_G^2 + V\hat{U}_k) \quad (19)$$

Let $A = C_2\eta_{\text{eff}}\sigma^2$ and $B' = C_4\eta_{\text{eff}}\tau V$. Since $\sum p_k B_G^2 = B_G^2$ (constant), minimizing $J(p)$ is equivalent to minimizing:

$$J'(p) = A\sum_{k=1}^{K}\frac{p_k^2}{|\mathcal{D}_k|/|\mathcal{D}|} + B'\sum_{k=1}^{K}p_k\hat{U}_k \quad (20)$$

subject to $\sum p_k = 1, p_k \geq 0$.

We use Lagrange multipliers. The Lagrangian $\mathcal{L}$ (ignoring $p_k \geq 0$ initially) is:

$$\mathcal{L}(p, \mu) = J'(p) - \mu\left(\sum_{k=1}^{K}p_k - 1\right) \quad (21)$$

Setting the partial derivative w.r.t. $p_k$ to zero:

$$\frac{\partial\mathcal{L}}{\partial p_k} = \frac{\partial J'}{\partial p_k} - \mu = 0 \quad (22)$$

$$\frac{\partial J'}{\partial p_k} = 2A\frac{p_k}{|\mathcal{D}_k|/|\mathcal{D}|} + B'\hat{U}_k \quad (23)$$

So, $2A\frac{p_k}{|\mathcal{D}_k|/|\mathcal{D}|} + B'\hat{U}_k - \mu = 0$. Solving for $p_k$ gives the optimal $p_k^*$ (before normalization and projection):

$$p_k^* = \frac{|\mathcal{D}_k|/|\mathcal{D}|}{2A}(\mu - B'\hat{U}_k) \quad (24)$$

Since $A = C_2\eta_{\text{eff}}\sigma^2 > 0$ and $B' = C_4\eta_{\text{eff}}\tau V \geq 0$ (assuming $V \geq 0$, consistent with Assumption 4.4 using V), this shows $p_k^*$ is proportional to $|\mathcal{D}_k|/|\mathcal{D}|$ and decreases linearly with $\hat{U}_k$.

This matches the conceptual form in Proposition 4.6: $p_k^* \propto \omega_k \times (\kappa - \zeta \times \hat{U}_k)$ where $\omega_k \propto |\mathcal{D}_k|/|\mathcal{D}|$, $\kappa$ relates to $\mu/(2A)$, and $\zeta = B'/(2A) = \frac{C_4\tau V}{2C_2\sigma^2}$. Since $C_2, C_4, \tau, V, \sigma^2$ are non-negative, $\zeta \geq 0$. If $V > 0$, then $\zeta > 0$, confirming the inverse relationship. The Lagrange multiplier $\mu$ is determined by the constraint $\sum p_k^* = 1$. The non-negativity constraint $p_k \geq 0$ might require projecting this solution onto the probability simplex if $\mu - B'\hat{U}_k$ becomes negative for some $k$.

# I  UNCERTAINTY AND MODEL PERFORMANCE IN ABLATION EXPERIMENTS

We selected FedAvg and UFL ($|\mathcal{I}|/|\mathcal{D}_k| = 5\%$) to investigate the impact of uncertainty on training performance. As shown in Table 4, datasets are classified into different difficulties. The average uncertainty is calculated by averaging the total uncertainty of high-uncertainty samples across all clients.

Table 4: Task Difficulty

| DATASETS | CLASSES | IMAGE TYPE | TASK DIFFICULTY |
|---|---|---|---|
| CIFAR-10 | 10 | COLOR | MEDIUM |
| SVHN | 10 | COLOR | MEDIUM |
| MNIST | 10 | GRAY | LOW |
| FASHION-MNIST | 10 | GRAY | LOW |
| CIFAR-100 | 100 | COLOR | HIGH |
| TINYIMAGENET-200 | 200 | COLOR | HIGH |

As shown in Figure 7, uncertainty varies across different tasks. In low-difficulty tasks, FedAvg already achieves excellent performance, leaving limited room for improvement. In Figure 7 (a) and (d), our method demonstrates improvement and can accelerate algorithm convergence. In high-difficulty tasks, UFL exhibits outstanding performance and reduces the uncertainty of the global model. Figure 7 illustrates that as the number of classes increases, the model's uncertainty over the dataset also rises. Therefore, in FL, more complex environments necessitate a greater focus on high-uncertainty samples. More details are in Appendix I.

To further confirm that focusing on 5% of the samples is sufficient to enhance the performance of the global model, we conducted a statistical analysis of the model's uncertainty and overall prediction accuracy. The results are shown in Figure 8. In the experiments, we selected 5%, 10%, 20%, 30%, 40%, and 50% of the samples as high-uncertainty samples to observe the impact of different proportions on model performance.

The results indicate that when 5% of the samples are selected as high-uncertainty samples, the model's prediction accuracy improves, and the model's uncertainty decreases. This suggests that a small number of high-quality high-uncertainty samples can effectively guide the model to focus on key features, enhancing its ability to recognize complex patterns.

However, when we select 30% of the samples as high-uncertainty samples, the accuracy is similar to that when selecting 5%, without further improvement. More importantly, as the proportion of high-uncertainty samples increases, the model's uncertainty begins to rise. This indicates that an excessive number of high-uncertainty samples may introduce noise, interfere with the model's learning process, and lead to instability in training and a decline in performance.

Furthermore, if the parameter $\alpha$ is set too high, it can lead to a decrease in performance due to the model over-focusing on high-uncertainty samples. This overemphasis may cause the model to neglect other important aspects of the data, resulting in suboptimal generalization.

Additionally, the number of N-Passes affects the model's performance. While varying the number of N-Passes does influence performance, the impact is not substantial. This suggests that the model is relatively robust to changes in the number of N-Passes, and other factors, such as sample selection strategy, play a more critical role in performance optimization.

Further analysis in ablation experiments shows the model's performance no longer improves with the increase in high-uncertainty samples. Instead, it may adversely affect the overall performance due to the degradation in sample quality. Therefore, an appropriate selection of high-uncertainty samples is crucial for optimizing the model training process.

In summary, this study demonstrates that controlling the proportion of high-uncertainty samples within a reasonable range (such as 5%) during the high-uncertainty sample selection process can effectively enhance model performance while avoiding the negative impacts of having too many high-uncertainty samples. This provides important guidance for selecting high-uncertainty samples in practical applications and emphasizes the critical role of sample selection strategies in model optimization.

# J  DISCUSSION ON THEORETICAL ANALYSIS

Our theoretical analysis provides a foundation for UFL grounded in standard FL convergence concepts while incorporating the novel aspect of MC Dropout-based uncertainty. By replacing the assumption on gradient

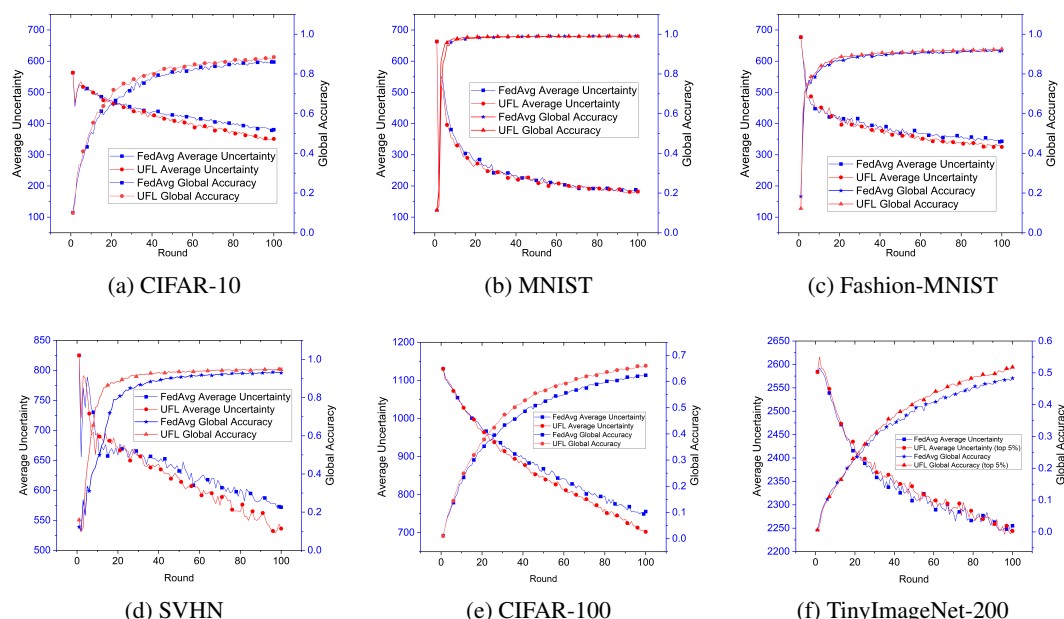

(a) CIFAR-10  (b) MNIST  (c) Fashion-MNIST

(d) SVHN  (e) CIFAR-100  (f) TinyImageNet-200

Figure 7: The relationship between uncertainty and the training process: UFL ($\mathcal{I} = 5\%$) vs. FedAvg. This experiment demonstrates that focusing on high-uncertainty samples leads to performance improvements, especially in more challenging scenarios ($C$=100, $C$=200).

norms with a more standard assumption linking client uncertainty $\hat{U}_k$ to gradient divergence (Assumption 4.4), we establish a clearer theoretical pathway.

The derived convergence bound (Theorem 4.5) explicitly shows how client uncertainty, through its contribution to gradient divergence, can potentially slow down convergence. Crucially, our analysis demonstrates that the optimal strategy to mitigate this effect involves adjusting aggregation weights (Proposition 4.6). Specifically, clients with higher uncertainty $\hat{U}_k$, which contribute more to the harmful divergence term in the bound, should receive lower aggregation weights $p_k^*$. This provides rigorous theoretical backing for the empirically successful U-Agg algorithm, which implements exactly this principle.

Compared to methods like FEDDISCO (Ye et al., 2023b), which use dataset-level distribution discrepancy measures, UFL leverages fine-grained, sample-level Bayesian uncertainty estimates. This potentially allows UFL to capture heterogeneity arising from difficult samples even when overall class distributions might appear similar. Our analysis suggests that using U-Agg weights (approximating $p_k^*$) can lead to a tighter convergence bound compared to standard FedAvg ($p_k = |D_k|/|\mathcal{D}|$), implying potentially faster convergence or convergence to a better solution, especially in settings with uncertainty-driven heterogeneity.

The UFL framework combines two synergistic components: 1. **Local Training**: The uncertainty estimation and loss weighting (using $\lambda_i$) focus on mitigating the impact of high-uncertainty samples locally, potentially reducing the client's inherent uncertainty $U_k$ over time. 2. **Global Aggregation (U-Agg)**: Leverages the current uncertainty information ($\hat{U}_k$) to perform robust aggregation, down-weighting clients that currently introduce higher divergence.

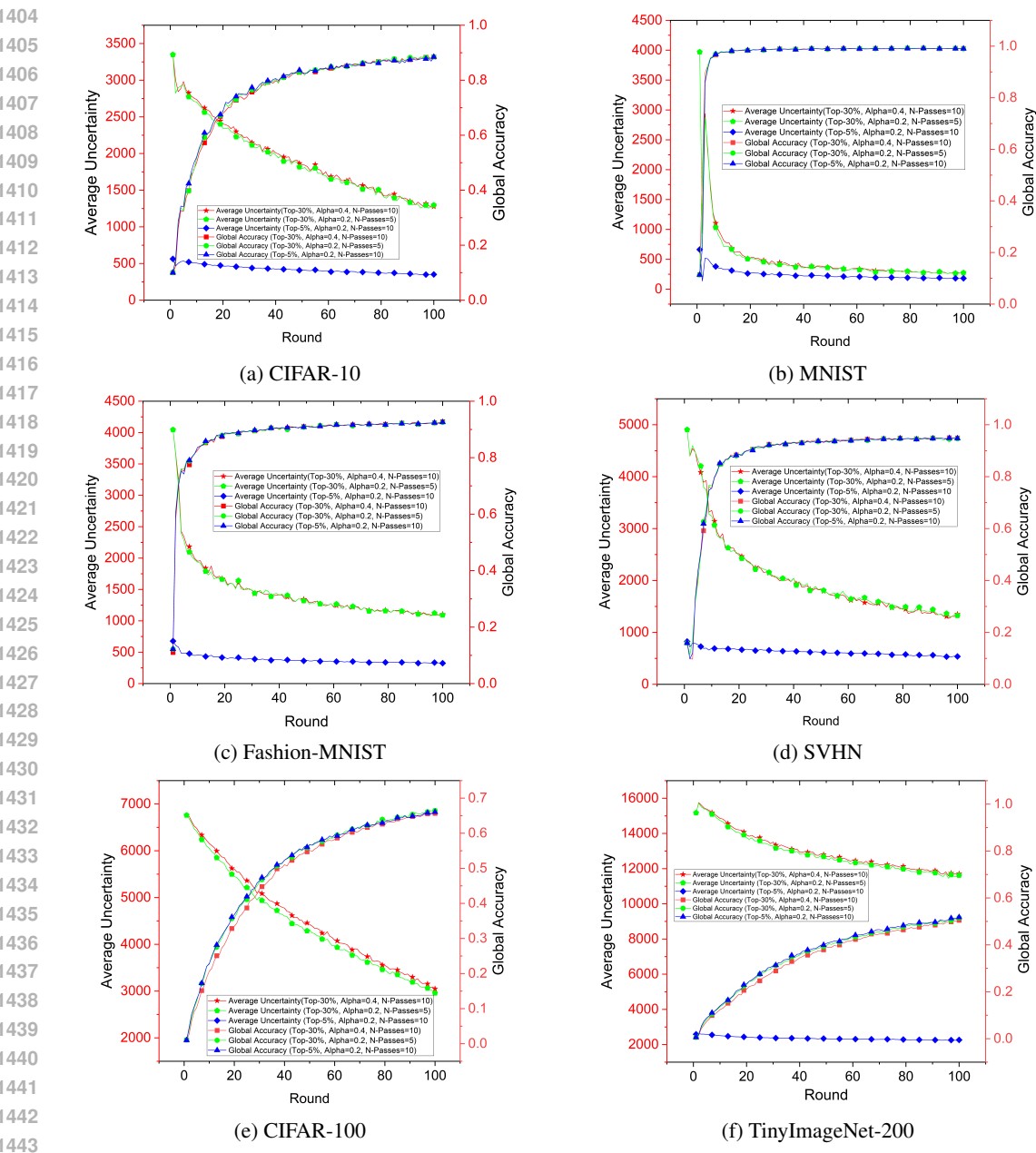

Figure 8: The relationship between uncertainty and the training process of ablation experiments

## K    MITIGATION OF HIGH-UNCERTAINTY IN FL

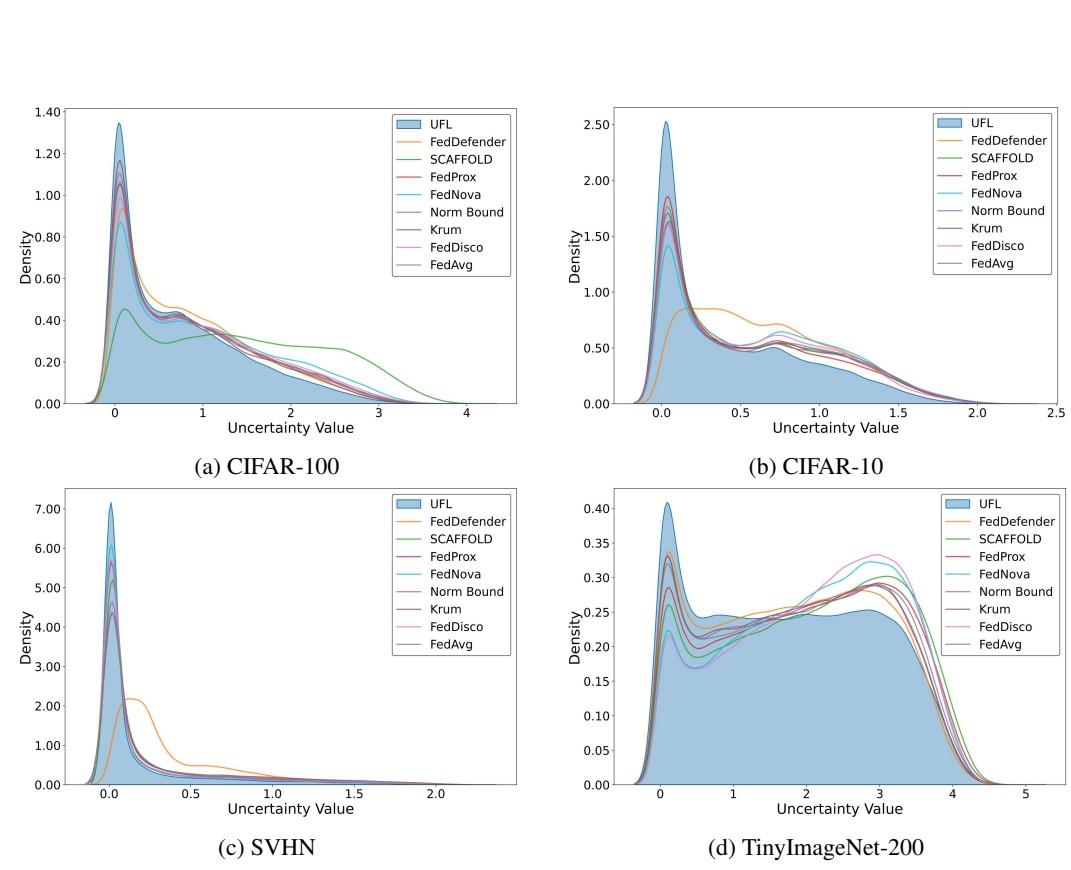

(a) CIFAR-100

(b) CIFAR-10

(c) SVHN

(d) TinyImageNet-200

Figure 9: Comparisons of the long tail effect.

