# OpenReview forum: "UFL: Uncertainty-Driven Federated Learning"
_ICLR.cc/2026/Conference — ICLR 2026 Conference Withdrawn Submission_

### Official Review · Reviewer_6KP4 · 2025-10-31

**Soundness:** 2
**Presentation:** 2
**Contribution:** 2
**Rating:** 2
**Confidence:** 4

**Summary:**

The paper introduces uncertainty aware FL, a framework designed to tackle the heterogeneity challenge in FL. The work is motivated from the observation that FL exhibits a long tail of sample uncertainty which means the model is less confident in the predictions unlike the centralized setting. The work addresses this by modifying both the local training and the aggregation mechanism. During local training the monte carlo dropout is used to estimate the samples level uncertainty and the samples are reweighted according to this uncertainty for the next round of training. For aggregation, the server down weights the parameter weights of the clients where the uncertainty is higher.

**Strengths:**

1. The paper is well written and easy to read.
2. The integration of the sample level uncertainty in local training and server aggregation is a nice idea is elegant.
3. Due to its generality, the idea can be utilized with many existing FL algorithms.

**Weaknesses:**

1. All the experimental results show very minor improvements over existing baselines, therefore the actual impact of the idea is not clear, especially given the fact that applying MC dropout during local training incurs significantly more costs for clients.\
2. The authors bring up sample level uncertainty to be intrinsic to FL, but it is only shown empirically, discussion around how and why this occurs would enhance the motivation of this work and may bring up new insights to develop a stronger algorithm.

**Questions:**

1. In different FL setups, how is the uncertainty distributed across clients? And how does that distribution affect the performance of the model?
2. The experiments section mention that each round involves only 1 epoch local training, does having more epochs worsen the performance?
3. How does the performance compare in IID FL setting?

---

> ### Author Response · Authors · 2025-11-26
>
> > (Q4-1) The improvements over baselines are minor, and MC dropout increases local cost.
>
> **Response to (Q4-1)**:
> We appreciate the reviewer’s comment. While the absolute accuracy gain of UFL is not large on some datasets, its significance lies in achieving consistent and stable improvements across strong baselines such as FedAvg, FedProx, SCAFFOLD, and FedNova. These methods are specifically designed to handle heterogeneous FL, and yet UFL still shows steady gains across different non-IID settings (CIFAR-10/100, TinyImageNet-200), indicating that it is not tailored to weak or specific conditions. More importantly, UFL effectively mitigates the “high-uncertainty long-tail” phenomenon, as observed in our empirical analysis, by stabilizing performance on structurally hard samples. This effect can be clearly seen in the KDE and uncertainty distribution plots in the appendix. Therefore, even if the accuracy improvement seems moderate, the method’s impact lies in its consistent robustness and ability to reduce instability in challenging non-IID environments.
>
> > (Q4-2) The use of MC dropout adds cost; please clarify feasibility.
>
> **Response to (Q4-2)**:  Thanks for your comments.
> We acknowledge the concern about additional cost and have carefully controlled the overhead. In our implementation, the number of MC dropout samples is kept small, and uncertainty estimation is performed only for a fixed subset of candidate samples in each round rather than for all samples. This keeps the computational increase moderate and confined to the training phase, with no effect on inference. The actual training time remains within a reasonable range. On CIFAR-10 with 10 clients, for example, the average time per round is as follows:
>
> | Method | Time (min) | Accuracy (%) |
> |--------|-----------:|-------------:|
> | **Ours** | **≈3** | **88.54** |
> | FedDefender | ≈6 | 87.95 |
> | FedProx | ≈2 | 86.44 |
> | FedAvg | ≈2 | 86.02 |
> | FedNova | ≈1 | 85.68 |
> | FedDisco | ≈1 | 85.54 |
> | Norm Bound | ≈2 | 85.47 |
> | Krum | ≈5 | 85.00 |
>
> These results show that the overhead of UFL is modest and remains acceptable compared with its stability and accuracy benefits.
>
> > (Q4-3) How is uncertainty distributed across clients, and how does it affect performance?
>
> **Response to (Q4-3)**:  Thanks for your comments.
> As shown in Section “Empirical Observations,” the distribution of uncertainty differs notably between IID and non-IID settings. Under near-IID partitions, the average client uncertainty and the proportion of high-uncertainty samples are relatively balanced. In contrast, under non-IID settings, a small number of clients concentrate most of the high-uncertainty samples, forming a clear long-tail pattern. After applying UFL, this tail is significantly reduced, which demonstrates its ability to mitigate such structural imbalance. We will include quantitative analysis in the revision to show that the gains of UFL become more pronounced when uncertainty is highly uneven across clients.
>
> > (Q4-4) Does using more local epochs worsen performance?
>
> **Response to (Q4-4)**:  Thanks for your comments.
> We used one local epoch per round mainly to maintain comparability with prior FL studies and to reflect practical constraints in real-world FL systems, where edge devices often have limited computing and energy budgets. Increasing local epochs would lengthen latency and strain devices, which contradicts practical deployment constraints. Nevertheless, since UFL only introduces an uncertainty-driven reweighting mechanism without altering the optimizer or communication protocol, it can be naturally extended to multi-epoch settings. We plan to add ablation studies in the revision to further analyze performance differences under various local training intensities.
>
> > (Q4-5) How does the method perform under IID settings?
>
> **Response to (Q4-5)**:  Thanks for your comments.
> Our primary focus is on non-IID scenarios, which represent the most challenging and realistic FL conditions where traditional methods suffer from degradation. The motivation and design of UFL target the instability arising from heterogeneous data and high-uncertainty samples. In near-IID conditions, uncertainty differences between clients diminish, and the improvement margin over FedAvg becomes smaller but remains stable. We agree that the current version lacks complete IID results and will add experiments and discussions in the revised manuscript to provide a more comprehensive evaluation across different data partition settings.

---

### Official Review · Reviewer_6znx · 2025-10-31

**Soundness:** 1
**Presentation:** 2
**Contribution:** 1
**Rating:** 2
**Confidence:** 5

**Summary:**

This manuscript is motivated by the so-called uncertainty, which is exactly the confidence values on ground truth classes. It claims that the "uncertainty" stems from the framework of FL itself, which is actually client drift during local training. The non-IID settings is actually class-wise uniform. Motivated by "sample-wise long-tail uncertainty", this paper propose to weight the aggregation ratio of global models.

**Strengths:**

- Simple method and easy to read.
- Clear convergence bound of a non-convex classical problem.

**Weaknesses:**

- The bad performance is the cause of uncertainty, not the other way around. Notice that the one-hot label already filtered other information than the ground truth. The metric is highly related to ground truth labels, where a better performance leads smaller uncertainty, i.e., the confidence values (probability logits) on ground truth classes. The main claim is invalid, because any models of bad performance could lead to the so-called "uncertainty". Under supervised testing data, a good model is only good in one way, while a bad model can be bad in a thousand strange and bizarre ways. This is why it seems like logarithm long-tail of predicted probabilities on ground truth classes.
- The theoretical analysis is a classic analytical framework on convergence rate of non-convex problem. The only uncertainty-aware term is B_{G}^{2}+V \hat{U}_{k} in Assumption 4.4, which is an upper bound of client-wise gradient variance. Not discussion about whether this assumptions holds at all. The related results is the 4-th term of the component B_{G}^{2}+V\hat{U}_{k}, the unchanged upper bound of classic clien-wise gradient variance!
- Thus, the no novel and dived-into empirical and theoretical insights are provided by this paper.

**Questions:**

Could your provide a correlation between performance (e.g., accuracy) and uncertainty on varied settings besides class-partition CIFAR-10?

---

> ### Author Response · Authors · 2025-11-26
>
> > (Q3-1) The bad performance is the cause of uncertainty, not the other way around.....
>
> Response to (Q3-1):
> Thank you for the comment. We do not claim that “uncertainty causes bad performance.” Our point is that, in non-IID federated learning, sample difficulty and client heterogeneity produce a clear long-tail uncertainty pattern, and using this signal during training and aggregation is more informative than relying only on sample counts or empirical loss. We estimate predictive uncertainty via MC Dropout,
> $$
> u_i = -\sum_{c=1}^C y_{i,c} \log \bar{p}_{i,c},
> $$
> and agree that for a fixed model, $u_i$ correlates with correctness. However, even at similar accuracy, non-IID FL shows a much heavier long tail than centralized/IID training, and high-uncertainty samples cluster near hard decision regions, indicating structural links to distribution shift rather than simple undertraining. Our method uses this signal: locally by reweighting high-uncertainty samples, and globally by uncertainty-based aggregation weights. Experiments (Figure 7) show consistent improvements over standard supervised training plus FedAvg.
>
> ---
>
> > (Q3-2) The theoretical analysis is a classic analytical framework on convergence rate of non-convex problem. ...
>
> Response to (Q3-2):
> Thank you for the comment. While we follow the standard non-convex FL framework, we do not reuse existing results. The key is incorporating client uncertainty into the chain “gradient divergence → convergence bound → aggregation weights.” Classical FL assumes
> $$
> \mathbb{E}\|\nabla F_k-\nabla F\|^2 \le B_G^2,
> $$
> where $B_G^2$ is a constant independent of data distribution. Our Assumption 4.4 replaces this with
>
> $$\mathbb{E}\|\nabla F_k(\theta)-\nabla F(\theta)\|^2 \le B_G^2 + V\hat{U}_k,$$
>
> making divergence depend on each client’s uncertainty. This turns the heterogeneity term into
> $
> C_4\,\eta_{\mathrm{eff}}\tau \sum_k p_k(B_G^2+V\hat{U}_k),
> $
> so the bound explicitly depends on $\sum_k p_k\hat{U}_k$. Optimizing over $p_k$ yields
> $$
> p_k^\star \propto |D_k|(\kappa-\zeta\hat{U}_k),
> $$
> showing why down-weighting high-uncertainty clients improves the bound. This uncertainty-dependent structure is the core difference from classical analyses.
>
> ---
>
> > (Q3-3) Thus, no novel and dived-into empirical and theoretical insights are provided by this paper.
>
> Response to (Q3-3):
> We understand the concern. Assumption 4.4 was explained briefly in the main text, but empirical evidence in the Appendix shows that clients with more high-uncertainty samples indeed have gradients deviating further from the global gradient. We will connect these results more clearly to the assumption. Our contribution is not a new optimization framework, but embedding an uncertainty-modulated divergence assumption into existing FL theory, deriving bounds containing $\sum_k p_k\hat{U}_k$, and designing justified uncertainty-based aggregation weights rather than heuristic tuning.
>
> ---
>
> > (Q3-4) Could your provide a correlation between performance (e.g., accuracy) and uncertainty on varied settings besides class-partition CIFAR-10?
>
> Response to (Q3-4):
> Thank you for the suggestion. Figure 7 already reports accuracy–uncertainty behavior on CIFAR-10, CIFAR-100, TinyImageNet, SVHN, MNIST, and Fashion-MNIST. The correlation persists across these datasets, especially in harder tasks with larger label spaces, where our method also provides larger gains, supporting its generality beyond CIFAR-10.

---

### Official Review · Reviewer_vyaw · 2025-11-01

**Soundness:** 2
**Presentation:** 3
**Contribution:** 2
**Rating:** 4
**Confidence:** 4

**Summary:**

This paper proposes UFL, a federated learning framework designed to address sample-level uncertainty. Specifically, UFL applies MC dropout to estimate sample-wise uncertainty, assigning higher weights to high-uncertainty samples during local client training. Furthermore, it utilizes the accumulated uncertainty across clients to adjust the global aggregation weights, down-weighting high-uncertainty local models and emphasizing low-uncertainty ones. Experiments demonstrate improvements over existing methods.

**Strengths:**

- The overall presentation is clear and well-organized.
- The paper is easy to follow and provides open-source code.
- The paper presents theoretical analyses to demonstrate that convergence is affected by client uncertainty.
- The paper provides comprehensive visualizations that effectively illustrate the proposed uncertainty phenomenon in federated learning.

**Weaknesses:**

This paper is a promising work but requires major revision before publication:
1. The uncertainty phenomenon is not well-explained. For instance:
- It’s unclear why the paper uses the uncertainty distributions at 75% epochs instead of others. IMHO, the distributions at 100% epochs show little to no long-tail behavior.
- It’s also unclear how the hard/easy samples are computed and identified in Figure 1 (c-e).
- The experimental settings of Figure 1 are missing, e.g., the number of clients and the degree of heterogeneity.
These missing details make it difficult to fully understand the uncertainty phenomenon.
2. The paper lacks a clear interpretation of how this long-tail uncertainty affects the federated learning system. A more detailed explanation (ideally with empirical analysis) would be much better.
3. The experimental evaluation is not comprehensive enough:
- The compared baselines are relatively old, and more recent approaches should be included.
- The ablation study only focuses on parameter sensitivity in UFL, missing the key ablations on the proposed components (i.e., the re-weighting and aggregation components).

**Questions:**

Please refer to the weaknesses above.

---

> ### Author Response · Authors · 2025-11-26
>
> > (Q2-1) 1.1 It’s unclear why ....
>
> **Response to (Q2-1)**: Thank you for your comment. We aim to show the long-tail uncertainty that remains in the mid-to-late training stages. Early in training, uncertainty is uniformly high and noisy, making long-tail patterns uninformative. Near full convergence, most samples are well fitted and the tail becomes visually compressed. Thus we choose the 75% training point as a clearer stage where the remaining high-uncertainty tail is still visible. Additional visualizations are provided in Figure 6 and Figure 7 of the Appendix.
>
> > (Q2-2) 1.2 It’s also ...
>
> **Response to (Q2-2)**: Thank you for your comment. To estimate hard and easy samples, we first compute the uncertainty of each sample using MC Dropout, then sort all samples by their uncertainty values. Samples in the lower part of the distribution are regarded as easy, while those in the higher part are regarded as hard. In our implementation, we simply treat the top 30% highest-uncertainty samples as hard samples.
>
> > (Q2-3) 1.3 The experimental ....
>
> **Response to (Q2-3)**: Thank you for pointing this out. This was an oversight in our writing. **Figure 1** adopts settings almost identical to the main experiments described later, using the **CIFAR-10** dataset. Other parameters, such as the number of clients, Non-IID partition methods and parameters, and local training hyperparameters, remain the same. We will supplement this relevant information in the revised version.
>
> > (Q2-4) The paper lacks a...
>
> **Response to (Q2-4)**: Thank you for your comment. We agree with your point that simply presenting the long-tail shape is not sufficient, and that its impact on federated training needs to be explained through empirical evidence. Our additional analyses show the following.
> 1. At the client level, when clients are grouped by their average uncertainty, those with higher uncertainty generally exhibit larger gradient norms and a greater deviation from the global gradient direction.
> 2. At the sample level, when samples are grouped by uncertainty, high-uncertainty samples are more likely to be misclassified or to stay near the decision boundary throughout training.
>
> In FedAvg, these factors introduce substantial volatility into the aggregated updates, often causing oscillation rather than smooth convergence in the later stages.
>
> > (Q2-5) The compared baselines are .....
>
> **Response to (Q2-5)**: Thank you for your comments. We would like to clarify that the current version already includes recent baselines such as FedDefender (2023) and FedDisco (2024), so our evaluation is not limited to early Federated Learning methods.
>
> We also conducted new experiments for [1] and [2].  Since the experiment for [2] is still ongoing and its data partitioning differs considerably from the standard non-IID setting, a direct comparison with the CLIP2FL results is not appropriate at this stage. We therefore compare only with [1]. As the authors’ code is not publicly available, we rely on the results reported in their paper. The corresponding details are summarized in the following table.
>
> **Table: Prediction accuracy under Dirichlet $\beta = 0.1$ non-IID distribution**
>
> | **Method**     | **CIFAR-10 (%)** | **CIFAR-100 (%)** |
> |----------------|------------------|-------------------|
> | FedAF [1]      | 69.11            | 50.61             |
> | UFL + Uagg     | 80.38            | 60.77             |
>
> [1] An aggregation-free federated learning for tackling data heterogeneity
> [2]  Clip-guided federated learning on heterogeneity and long-tailed data
>
> >(Q2-6) The ablation study ....
>
> **Response to (Q2-6)**: Thanks for your comments.
> We computed local uncertainties using MC Dropout without modifying the local loss function, and these uncertainties were used only for determining the aggregation weights. The results on CIFAR-10 and CIFAR-100 are reported in Table 2.
>
> **Table 2. Ablation study on the effect of the local re-weighting strategy**
>
> | **Method**         | **CIFAR‑10 (%)** | **CIFAR‑100 (%)** |
> |-|-|-|
> | Local SGD + U‑Agg  | 88.04            | 65.81             |
> | **UFL + U‑Agg**    | **88.54**        | **66.56**         |
>
> We implemented the FedAdam aggregation rule in our code and evaluated it on CIFAR-10 and CIFAR-100 due to limited time and computational resources. FedAdam reached 73.11% accuracy on CIFAR-10 and 25.80% on CIFAR-100, while the results reported in S. Reddi et al., “Adaptive Federated Optimization” are 77.40% and 52.50%. Our method achieves 88.54% on CIFAR-10 and 66.56% on CIFAR-100, as shown in Table 1.
>
>
> **Table 1. Accuracy comparison with FedAdam**
>
> | **Method**                    | **CIFAR‑10 (%)** | **CIFAR‑100 (%)** |
> |-|-|-|
> | FedAdam (our reproduction)   | 73.11            | 25.80             |
> | FedAdam (Reddi *et al.*, 2020) | 77.40          | 52.50             |
> | **UFL + U‑Agg (ours)**       | **88.54**        | **66.56**         |

---

### Official Review · Reviewer_yjLu · 2025-11-04

**Soundness:** 2
**Presentation:** 2
**Contribution:** 2
**Rating:** 2
**Confidence:** 5

**Summary:**

The authors handle data heterogeneity (DH) in federated settings and approximates the sample-level uncertainty via Monte Carlo Dropout (Gal & Ghahramani, 2016). The authors have shown some convergence results.

**Strengths:**

Addressing sample-level uncertainty is a major and important problem.

**Weaknesses:**

- Robust aggregation schemes against adversarial updates/nodes has nothing with handling distribution shifts and data heterogeneity.These are two distinct problems with different objectives and solutions.

- Assumption 4.4 does not capture any meaningful distribution shift. The upper-bound can be very large, which makes the bound impractical. Convergence results in Theorem 4.5 do not provide any insight into handling data heterogeneity under a meaningful and practical setting.

- Baselines in Figure 2 do not handle data heterogeneity in terms of distribution shifts. Some important baselines are missed:

[4] A. Ramezani-Kebrya, F. Liu, T. Pethick, G. Chrysos, and V. Cevher. Federated learning under covariate shifts with generalization guarantees. TMLR 2023.

[5] Z. Wu, C. Choi, X. Cao, V. Cevher, and A. Ramezani-Kebrya. Addressing label shift in distributed learning via entropy regularization. ICLR 2025.

**Questions:**

Please address the weaknesses above.

---

> ### Author Response · Authors · 2025-11-26
>
> > **(Q1-1)** Robust aggregation schemes against adversarial updates/nodes has nothing with handling distribution shifts and data heterogeneity. These are two distinct problems with different objectives and solutions.
>
>
> **Response to Q1-1.**
> Thank you for the comment. Our intention was not to treat Byzantine-robust aggregation as a “standard solution” for distribution shift. We only pointed out that some robust schemes (e.g., trimmed mean, Krum) incidentally suppress extreme gradients that arise under strong data heterogeneity, and thus act as a coarse mechanism for mitigating DH in practice.
> U-Agg follows the idea of suppressing harmful updates through weighting, but **assumes no adversarial clients**. High-uncertainty clients are regarded as unstable contributors during optimization, not as malicious nodes.
> In the revision, we will clearly separate the two research lines:  (1) Byzantine/adversarial-robust FL (malicious clients), and   (2) Statistical DH/distribution shift (non-malicious, distributionally different clients).   U-Agg belongs entirely to (2), and its design and convergence analysis are built purely on non-adversarial DH.
>
> ---
> > **(Q1-2) 2. Assumption 4.4 does not capture any meaningful distribution shift. The upper-bound can be very large, which makes the bound impractical. Convergence results in Theorem 4.5 do not provide any insight into handling data heterogeneity under a meaningful and practical setting.**
>
> **Response to Q1-2.**
> Thank you for the insightful suggestion.
> Regarding Assumption 4.4, we assume:
> $
> \mathbb{E}\|\nabla F_k(\theta_k)-\nabla F(\theta)\|^2 \le B_G^2 + V\,\hat{U}_k ,
> $
> where $\hat{U}_k$ is the aggregated uncertainty of client $k$. Classical FL theory typically uses a client-independent bound $\le B_G^2$. Our intention is not to model a specific covariate/label shift, but to express heterogeneity via gradient divergence and link the *extra* divergence to an observable quantity—client uncertainty.
>
> This assumption reflects an empirical fact: clients with covariate shift, label imbalance, or long-tail classes often yield higher prediction uncertainty and more unstable gradients. Assumption 4.4 formalizes this monotonic relation without requiring an explicit shift model.
>
> We agree constants such as $B_G^2$ and $V$ may be large in worst cases, as is standard in FL analysis. The purpose of the assumption is structural, not numerical tightness. In Theorem 4.5, it appears in:
>
> 1. The heterogeneity term
> $
> C_2\,\eta_{\mathrm{eff}}\tau \sum_k p_k(B_G^2 + V\hat{U}_k),
> $
> 2. The optimal aggregation direction
> $
> p_k^\star \propto |D_k|(\kappa - \zeta \hat{U}_k),
> $
> showing that down-weighting high-uncertainty clients directly reduces the deterioration term.
>
> Thus, Theorem 4.5 provides two insights:
> (i) high-uncertainty clients worsen the convergence bound through $\sum p_k\hat{U}_k$;
> (ii) making $p_k$ negatively correlated with $\hat{U}_k$ theoretically improves the bound.
> We will clarify this structural role in the revision.
>
> ---
>
> > **(Q1-3) 3. Baselines in Figure 2 do not handle data heterogeneity in terms of distribution shifts. Some important baselines are missed:**
>
> **Response to Q1-3.**
> Thank you for highlighting these recent works. In this version, we compared mainly classical FL baselines (FedAvg, FedProx, SCAFFOLD, FedNova, etc.) and robust/regularized aggregation rules under the assumption that no explicit target distribution information (e.g., density ratios) is available, which matches the U-Agg setting.
>
> The suggested baselines are indeed important:
> - [4] provides covariate-shift generalization guarantees via density-ratio–based weighted ERM;
> - [5] handles label shift by estimating label-importance ratios with entropy regularization.
>
> These methods assume access to (or accurate estimation of) density ratios. In contrast, U-Agg requires no density-ratio estimation and relies solely on model prediction uncertainty to guide aggregation. Thus our current baseline selection focuses on methods operating under the same assumption of “no extra target distribution information.”
>
> In the revision, we will add a subsection comparing U-Agg with density-ratio–based FL (e.g., [4], [5]) and explain their complementary nature and include experiments with [4] and [5] if space and time permits.

---

### Note · Authors · 2025-12-01

I have read and agree with the venue's withdrawal policy on behalf of myself and my co-authors.